# Gradient-based Explanations for Deep Learning Survival Models

**Sophie Hanna Langbein** [* 1 2]  **Niklas Koenen** [* 1 2]  **Marvin N. Wright** [1 2 3]

## Abstract

Deep learning survival models often outperform classical methods in time-to-event predictions, particularly in personalized medicine, but their "black box" nature hinders broader adoption. We propose a framework for gradient-based explanation methods tailored to survival neural networks, extending their use beyond regression and classification. We analyze the implications of their theoretical assumptions for time-dependent explanations in the survival setting and propose effective visualizations incorporating the temporal dimension. Experiments on synthetic data show that gradient-based methods capture the magnitude and direction of local and global feature effects, including time dependencies. We introduce GradSHAP(t), a gradient-based counterpart to SurvSHAP(t), which outperforms SurvSHAP(t) and SurvLIME in a computational speed vs. accuracy trade-off. Finally, we apply these methods to medical data with multi-modal inputs, revealing relevant tabular features and visual patterns, as well as their temporal dynamics.

## 1. Introduction

As medical databases expand to include patients' detailed medical history and genetic information, healthcare is shifting from population-based models and traditional statistical approaches targeting the "average" patient to more complex personalized medicine, which tailors diagnoses to individual patient characteristics. Survival analysis is fundamental to medical data analysis, modeling time-to-event outcomes while accounting for censoring, enabling personalized risk predictions, and assessing treatment effects to advance clini-

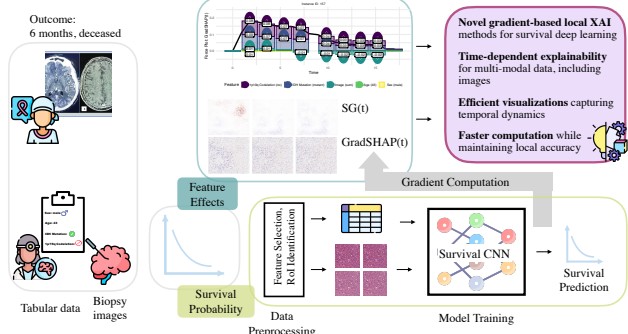

*Figure 1.* Overview of our workflow for generating time-dependent post-hoc explanations using gradient-based methods by the example of overall brain cancer survival prediction. The approach utilizes a survival deep learning model with multi-modal input data, providing insights into the temporal dynamics of feature effects through tailored visualizations for different feature types. (images: Flaticon.com)

cal research and evidence-based medicine. Deep learning methods hold significant potential for advancing survival analysis by framing pathogenic identification as a data-driven problem, uncovering correlations between patient profiles and disease phenotypes, and seamlessly learning from unstructured or high-dimensional data such as images, text, or omics, thereby revealing hidden and complex patterns undetectable by classical approaches (Zhang et al., 2019). While machine learning models show great promise for survival analysis, their inherent opacity raises legitimate concerns, as in fields like life sciences, interpretability is crucial to support sensitive decision-making, mitigate biases, promote equity, and ensure compliance with regulatory standards (Rahman & Purushotham, 2022; Vellido, 2020). In recent years, several post-hoc eXplainable AI (XAI) methods for population-wide (*global*) and individual (*local*) insights into the decision-making process of machine learning survival models have been proposed (Langbein et al., 2024). For personalized medicine, the local model-agnostic approaches SurvLIME (Kovalev et al., 2020), an extension of LIME (Ribeiro et al., 2016), and SurvSHAP(t) (Krzyziński et al., 2023), a generalization of SHapley Additive exPlanations (SHAP) (Lundberg & Lee, 2017), have prevailed. However, no methods specifically targeted at or practical to

*Equal contribution [1]Leibniz Institute for Prevention Research and Epidemiology – BIPS, Bremen, Germany [2]Faculty of Mathematics and Computer Science, University of Bremen, Germany [3]Department of Public Health, University of Copenhagen, Denmark. Correspondence to: Marvin N. Wright <wright@leibniz-bips.de>.

*Proceedings of the 42ⁿᵈ International Conference on Machine Learning*, Vancouver, Canada. PMLR 267, 2025. Copyright 2025 by the author(s).

survival deep learning models have been introduced.

In this paper, we introduce a formal framework generalizing gradient-based explanation methods for individual predictions (i.e., local) to survival neural networks (NNs), extending their applicability to functional form outcomes beyond traditional regression and classification tasks, and showcasing their applicability in multi-modal patient-level survival prediction.

**Contributions.** We extend a set of six representative gradient-based explanation methods (e.g., Saliency, IntegratedGradient, GradSHAP, and Gradient×Input) to time-to-event survival analysis, addressing a crucial gap in survival XAI research. Our main contributions are:

(1) We adapt and systematically assess **gradient-based XAI methods** in the context of **time-to-event functional outcomes**, analyzing both their theoretical assumptions and practical challenges, e.g., for the gradient calculation in deep survival NN models.

(2) We develop **visualization** and **interpretation techniques** for functional outputs tailored to the different gradient-based explanation methods. In doing so, we contribute to the ongoing debate and disagreement regarding gradient-based methods (Sturmfels et al., 2020; Krishna et al., 2024; Koenen & Wright, 2024b) by clarifying how implicit or explicit baselines in these methods influence survival explanations.

(3) We introduce **GradSHAP(t)**, a gradient-based, model-specific counterpart to SurvSHAP(t) for SHAP-like explanations. A quantitative comparison confirms that the gradient-based approach outperforms the sample-based version and SurvLIME, offering a better balance between computational speed and local accuracy.

(4) Using a **multi-modal real-world brain cancer survival example** with tabular and histopathological image inputs, we demonstrate that the methods can feasibly identify prediction-relevant features including their temporal dynamics and visual patterns (see Fig. 1).

(5) All methods and visualization tools are implemented in our **open-source R package** `survinng`[1], which supports `torch`-based survival models from `survivalmodels` (Sonabend, 2024) and PyTorch models trained in `pycox` (Kvamme et al., 2019).

## 2. Background

### 2.1. Survival Data

We are considering survival analysis as a supervised prediction task for the time-to-event distribution of a given dataset $\mathcal{D}$. We limit ourselves to the standard right-

___
[1]https://github.com/bips-hb/survinng

censored setting, in which the data consist of $n$ triplets, i.e., $\mathcal{D} = \left\{ (\boldsymbol{x}^{(i)}, y^{(i)}, \delta^{(i)}) \right\}_{i=1}^{n}$. The first component, $\boldsymbol{x}^{(i)} = (x_1^{(i)}, x_2^{(i)}, \ldots, x_p^{(i)}) \in \mathcal{X}$, represents the $p$-dimensional vector of predictive features for *individual $i$*, which may include data types commonly used in classical supervised learning, such as images or tabular data. The observed right-censored time $y^{(i)}$ is defined as the minimum of the event time $t^{(i)} \in \mathbb{R}_0^+$ and the censoring time $c^{(i)} \in \mathbb{R}_0^+$, i.e., $y^{(i)} = \min(t^{(i)}, c^{(i)})$. The binary event indicator $\delta^{(i)} \in \{0, 1\}$ takes the value 0 if the observation is censored ($t^{(i)} > c^{(i)}$) and 1 if the event occurs ($t^{(i)} < c^{(i)}$). For clarity, we will omit the superscript for an instance $i$ in the remaining text when it is not necessary.

### 2.2. Survival Distribution Representations

Key quantities to be modeled and predicted in survival analysis are distributional representations of the random variable $T$ on the time domain $\mathcal{T}$, typically a subset of $\mathbb{R}_0^+$. We generalize them as $f : \mathcal{X} \times \mathcal{T} \to \mathbb{R}$, a set of functions that map value combinations from the feature space $\mathcal{X}$ and the time space $\mathcal{T}$ to a one-dimensional outcome. The most popular representations are survival $S$, hazard $h$, and the cumulative hazard function $H$.

**Definition 2.1** (Survival Function). The *survival function* $S : \mathcal{X} \times \mathcal{T} \to [0, 1]$ describes the probability of the time-to-event (survival time) being greater than or equal to a specific time point $t \geq 0$ conditional on the observed features $\boldsymbol{x} \in \mathcal{X}$

$$S(t|\boldsymbol{x}) := \mathbb{P}(T \geq t|\mathbf{x}) = 1 - \mathbb{P}(T \leq t|\boldsymbol{x}) . \quad (1)$$

**Definition 2.2** (Hazard Function). The *hazard* or *risk function* $h : \mathcal{X} \times \mathcal{T} \to \mathbb{R}_0^+$ describes the instantaneous risk of occurrence of the event of interest in an infinitesimally small time interval $[t, t + \Delta t]$ for continuous $t \in \mathbb{R}_0^+$, given that it has not yet occurred before time $t$ and conditional on the observed features $\boldsymbol{x} \in \mathcal{X}$

$$h(t|\boldsymbol{x}) := \lim_{\Delta t \to 0} \frac{\mathbb{P}(t \leq T \leq t + \Delta t|T \geq t, \boldsymbol{x})}{\Delta t} \quad (2)$$

$$= -\frac{\mathrm{d}}{\mathrm{d}t} \log S(t|\boldsymbol{x}) . \quad (3)$$

**Definition 2.3** (Cumulative Hazard Function). The *cumulative hazard function* $H : \mathcal{X} \times \mathcal{T} \to \mathbb{R}_0^+$ describes the accumulated risk of experiencing the event of interest up to a specific time $t \in \mathbb{R}_0^+$ conditional on the observed features $\boldsymbol{x} \in \mathcal{X}$

$$H(t|\boldsymbol{x}) := \int_0^t h(u|\boldsymbol{x})du = -\log(S(t|\boldsymbol{x})) . \quad (4)$$

### 2.3. Related Work

Deep learning survival models often extend the Cox regression framework by using NNs to parameterize the log-risk

function, optimizing a Cox-based loss, i.e., the negative log-partial likelihood of the Cox model (Cox, 1972). Examples include *DeepSurv* (Katzman et al., 2018), which employs feedforward NNs to capture non-linear feature-hazard relationships while adhering to the *proportional hazards* (PH) assumption. It states that the hazard ratio between any two individuals remains constant over time, meaning the effect of features on the hazard function is multiplicative and does not vary with time. *CoxTime* (Kvamme et al., 2019) incorporates time-dependent feature effects by including time as an additional feature. Another category of survival NNs adapts discrete-time methods, treating time as discrete to leverage classification techniques. The most prominent model in this category is *DeepHit* (Lee et al., 2018), which directly models the joint distribution of survival times and event probabilities without assumptions about the stochastic process, allowing dynamic feature-risk relationships. Additional approaches include methods based on piecewise-exponential models, ordinary differential equations, and ranking techniques. For a comprehensive review, see Wiegrebe et al. (2024). To date, the interpretability of survival NNs has primarily been addressed through model-agnostic, post-hoc methods. Prominent local XAI techniques in this domain include SurvLIME and SurvSHAP(t). For a comprehensive review of interpretability methods in survival analysis, we refer to Langbein et al. (2024). So far, application-focused studies have employed existing model-specific XAI methods, such as simple gradients, to analyze deep learning survival models – primarily to identify important nodes or generate saliency maps for singular images (Mobadersany et al., 2018; Hao et al., 2019; Cho et al., 2023). However, to the best of our knowledge, no study has explicitly extended these methods to general time-dependent explainability approaches for survival NNs.

## 3. Taxonomy of Deep Survival Models

For the application of gradient-based feature attribution techniques, we categorize survival NNs based on two criteria: (1) supported **input feature modalities** and (2) their **prediction outcomes**.

**Prediction outcome.** In survival NNs, the prediction outcome refers to the network's output, $f : \mathcal{X} \times \mathcal{T} \rightarrow \mathcal{Y}$. These outcomes correspond to the distributional representations discussed in Sec. 2.2 as, e.g., the hazard function or survival function. Differentiating based on prediction outcome is critical, as it constitutes the quantities decomposed during attribution and capturing the effects of changes in input features.

**Input feature modalities.** Differentiation by feature modality is essential, as it affects how attribution values are visualized. The input modalities a network can process

depend on its architecture; while many survival DL models use feedforward NNs, other architectures such as convolutional NNs (CNNs), recurrent NNs, generative adversarial networks, and autoencoders are also employed (Wiegrebe et al., 2024). Below, we discuss the relevant feature modalities and their coverage in this work.

(1) *Time dependence*: Time dependence can be incorporated in two ways: 1) time-varying effects of time-constant features (TVE) or 2) time-varying features (TVF). The PH assumption simplifies the feature effect on the hazard scale to a one-dimensional setting $f : \mathcal{X} \rightarrow \mathcal{Y}$, similar to standard regression or classification. However, attribution values are generally not time-constant, even in PH models, when evaluated on survival or cumulative hazard scales, necessitating time-dependent feature attribution computation and visualization. TVF in time-to-event prediction are analogous to longitudinal input data and require specialized architectures, such as recurrent NNs, along with tailored interpretability methods (Ferreira et al., 2021).

(2) *Data modalities*: Mixed tabular data are one of the most common input types for survival analysis and many survival deep learning methods have been (first) developed for it (Katzman et al., 2018; Kvamme et al., 2019; Lee et al., 2018). High-dimensional (multi-)omics data are another popular input type for survival NNs, with many specialized NNs developed for this purpose (Ching et al., 2018; Hao et al., 2018; 2019). Conceptually, feature attribution techniques apply similarly to high- and low-dimensional inputs, as feature contributions are considered individually. Some omics-specific NNs assign biological meaning to network nodes, enabling feature attributions to quantify the impact of biologically significant quantities. Another key data modality for survival NNs is (often medical) image data, typically processed using convolutional NN architectures (Zhu et al., 2016; Mobadersany et al., 2018; Tang et al., 2019). Since many gradient-based feature attribution methods were originally designed for image data, their adaptation to survival models is straightforward for a single outcome time point, where traditional saliency maps (Simonyan et al., 2014) can be generated. Saliency across multiple time points can be represented using different maps, colors or visual markers in a single map. Other input formats, such as text data, are less frequently used for time-to-event predictions and are not covered in this work. In our experiments, we consider DeepSurv, CoxTime and DeepHit as a representational set of deep survival models.

## 4. Gradient-based Survival Explanations

Gradient-based feature attribution methods are a set of local model-specific XAI techniques that assign relevance scores to input features based on their contribution to the prediction of a NN. These methods efficiently leverage the automatic

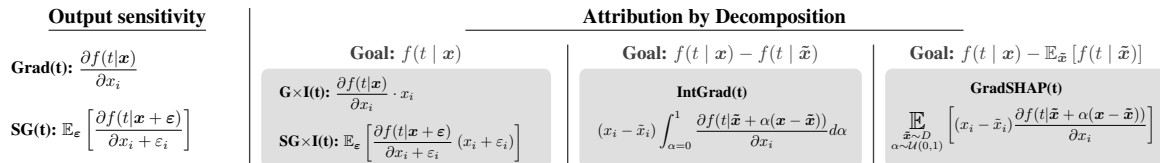

*Figure 2.* Mathematical representations of gradient-based feature attribution methods adapted to survival NNs. Each block corresponds to a different underlying objective. For example, in the case of feature-wise relevances $R_j^t$ obtained from G×I(t), the goal is to achieve a sum that equals $f(t|\boldsymbol{x})$, i.e., $\sum_{j=1}^p R_j^t = f(t|\boldsymbol{x})$.

differentiation capabilities inherent in modern deep learning libraries (Chollet et al., 2015; Paszke et al., 2019) to quantify how each input feature influences the model's output (Ancona et al., 2019; Koenen & Wright, 2024a). In their original formulation, these methods are defined for scalar outputs resulting in an attribution value $R_j$ for each feature $j$. However, in the survival context, the outcome is represented as a prediction vector for different time points, which complicates the computation and adds an additional dimension to the explanations. Instead of being applied to a single $f(\boldsymbol{x}) \in \mathbb{R}$, a survival XAI method is applied at each discretized time point $f(t_0|\boldsymbol{x}), \ldots, f(t_T|\boldsymbol{x})$, resulting in an attribution value $R_j^{t_k}$ for each feature $j$ and time point $t_k$, thus, an ensemble of explanations including their temporal interplay. Computationally, a straightforward implementation is not always feasible. For example, replicating a single instance across multiple time points, as required by CoxTime, violates the assumption of independence between samples and leads to unintended gradient accumulation. Therefore, in the following, we extend the most common gradient-based feature attribution methods to survival networks and thereby propose their time-dependent counterparts Grad(t), SG(t), G×I(t), IntGrad(t) and GradSHAP(t); see Fig. 2 for an overview of the proposed methods.

### 4.1. Output-Sensitivity Methods

Although they do not represent attributions in the classical sense[2], output-sensitive methods are often categorized as feature attribution. As pointed out in Koenen & Wright (2024b), methods in this category do not produce actual attributions but rather local importance tendencies.

**Grad(t).** The Gradient method, also known as Vanilla Gradient or Saliency Maps in the image domain, developed by Simonyan et al. (2014), is one of the earliest and most intuitive attribution methods in deep learning. For Grad(t) we compute relevance scores as the (absolute) partial derivatives of the target outcome $f(t|\boldsymbol{x})$ with respect to the corresponding input feature $\boldsymbol{x}_j$ at a particular time

---

[2]This refers to an (approximate) decomposition of the output, such as *local accuracy* (Lundberg & Lee, 2017) or *completeness* (Sundararajan et al., 2017).

point $t$, as shown in Fig. 2. This captures how sensitive the prediction is to changes in each feature at $t$ and, over all time points, how the relevance evolves over time.

**SG(t).** Smilkov et al. (2017) propose smoothed gradients (SmoothGrad) as an extension to reduce noise in the raw gradients. Equivalently, in SG(t), the expected values of the gradients are computed over random Gaussian perturbations of the input, $\varepsilon \sim \mathcal{N}(0, \sigma^2)$ (see Fig. 2). In practice, the expected value of the gradients is estimated as an average over $K$ samples, so that the accuracy of the estimation can be improved by larger values of $K$. The Gaussian standard deviation $\sigma$ controls the sharpness of the explanation and is mostly indirectly specified through a noise level $\sigma = \frac{\sigma}{x_{\max} - x_{\min}}$, determining the proportion of the total range of the input domain that is covered by the standard deviation $\sigma$. Note, that in the survival setting, we can also perturb inputs over different time points (e.g., for CoxTime) to capture the temporal dynamics of feature effects.

### 4.2. Attribution-based Methods

Feature attribution methods typically aim to approximate a decomposition of some prediction-based quantity of a model into feature-wise additive contributions (Ancona et al., 2019; Shrikumar et al., 2017; Sundararajan et al., 2017). The quantity depends on the method and is often the model's output or a baseline-adjusted version. This property is commonly referred to as *local accuracy*.

**G×I(t).** A simple and computationally efficient approach for decomposing a prediction is to multiply the gradient by the corresponding feature value, as proposed by Shrikumar et al. (2017) known as Gradient×Input. Our survival adaption is denoted as G×I(t) in Fig. 2. Mathematically, this method is based on a first-order Taylor expansion at the implicitly set reference value zero, effectively providing a linear approximation of the model's output (Ancona et al., 2019; Montavon et al., 2017). However, since survival functions are inherently highly non-linear, this approach often leads to limitations in the accuracy of the decomposition.

**IntGrad(t).** The Integrated Gradients (IntGrad) method by Sundararajan et al. (2017) attributes the contribution of each feature by comparing a model's prediction for a given input $x$ to that of a baseline instance $\tilde{x}$. This method satisfies several desirable properties, including completeness, sensitivity, linearity, and implementation invariance, making it a widely accepted approach for feature attribution. Int-Grad computes contributions by integrating gradients along a path from the baseline to the input, typically following a straight line. In practice, this integral is approximated by averaging gradients at discrete intervals. The generalized IntGrad(t) results in a decomposition of the targeted curve of $x$ minus the baseline curve of $\tilde{x}$, e.g., for the survival curve, $\hat{S}(t|x) - \hat{S}(t|\tilde{x})$ for each time point $t$. Typical baseline values are zeros or the feature mean representing the "average patient". However, especially with non-linear or multi-modal distributions, careful selection of the reference value is required. It can be shown that for a non-negatively homogeneous model and $\tilde{x} = 0$, IntGrad is equivalent to Grad×Input (Hesse et al., 2021).

**GradSHAP(t).** In many cases, it is most meaningful for the baseline value $\tilde{x}$ to conceptually reflect the complete absence of features. However, for tabular data, choosing an adequate baseline can be challenging since a zero or mean value does not necessarily coincide with feature absence and may be out-of-distribution. In survival analysis, zero as a baseline can misrepresent missingness because it often carries specific clinical meaning (e.g., a zero lab value might imply a specific medical condition), leading to biased explanations if not carefully chosen. Our time-dependent extension, GradSHAP(t), of the GradSHAP method (Lundberg & Lee, 2017; Erion et al., 2021) addresses this by taking the expectation of IntGrad(t) explanations evaluated at randomly drawn reference values from $\mathcal{D}$ instead of using a single potentially off-manifold baseline value (see Fig. 2). In survival analysis, using the expectation over a reference distribution can provide a generalized baseline, reflecting the "mean patient's survival time" and offering more stable and less biased comparisons. In practice, the expectation is estimated by Monte Carlo integration using the sample average of randomly drawn baseline values from $\mathcal{D}$ and parameterized points $\alpha \sim \mathcal{U}(0, 1)$ for the integration path. This results in a decomposition of $f(t|x) - \mathbb{E}_{\tilde{x}}[f(t|\tilde{x})]$ and describes a gradient-based approximation of SHAP values at selected time points.

# 5. Experiments

Unlike for the evaluation of ML models, there is no well-established framework for XAI evaluation due to the absence of a definitive ground truth for explanations and the reliance on an imprecise black-box model trained with imperfect data (Liu et al., 2021; Vilone & Longo, 2021; An-

toniadi et al., 2021). To ensure a comprehensive evaluation strategy, we conduct experiments on simulated and real data to answer the following research questions:

1. Can gradient-based methods correctly identify **time-(in)dependent effects** in different survival NNs?
2. How does GradSHAP(t) compare to SurvSHAP(t) and SurvLIME in terms of **local accuracy**, **computational speed** and **global feature rankings**?
3. How can gradient-based methods be feasibly leveraged to explain survival predictions based on **multi-modal input data**?

All experiments, including the figures, can be reproduced using our code on GitHub[3].

## 5.1. Experiments on Simulated Data

### 5.1.1. TIME-INDEPENDENT EFFECTS

**Setup.** In the first experiment, we generate synthetic data to demonstrate that gradient-based explanations can accurately capture time-independent local feature effects, provided the models correctly identify them. The data consist of $N = 10,000$ observations simulated from a standard Cox PH model. The baseline hazard function is monotonically increasing and modeled using a Weibull distribution with a shape parameter $\gamma$ of 2.5. The features include one "harmful" feature $x_1$ with a log hazard ratio of 1.7, one "protective" feature $x_2$ with a log hazard ratio of -2.4, and one feature with no effect on the hazard $x_3$. The maximum follow-up time is set to $t = 7$. More details on the simulation setting, training process, fitted models and selected observations can be found in Appendix A.1. For all experiments, we use our R package `survinng`[4].

**Results.** These first experiments aim to show how different gradient-based explanation methods identify the effects of time-independent features. For this purpose, we split the data into training (9,500 observations) and test set (500 observations) and fit a DeepSurv (Katzman et al., 2018), a CoxTime (Kvamme et al., 2019), and a DeepHit (Lee et al., 2018) model to the training set.

Grad(t) (Fig. A.2) and SG(t) (Fig. A.4) are output-sensitive methods; as such they indicate the models' sensitivity to feature changes rather than appropriately capturing local effects on the survival prediction. Therefore, the global ground-truth effects are accurately reconstructed, with $x_1$ having a negative, $x_2$ a stronger positive, and $x_3$ no substantial effect on survival over time across all models for both of the randomly chosen instances. In contrast, the attribution curves in Figures A.3, A.5 - A.8 capture feature-wise local effects.

---

[3]https://github.com/bips-hb/Survival-XAI-ICML
[4]https://github.com/bips-hb/survinng

G×I(t) uses simple scaling of the sensitivity to account for the magnitude of the feature's contribution to the prediction. Thus, despite a negative global ground-truth effect of $x_2$, since $x_2 < 0$ for the 13th observation, its attribution curve only takes positive values as highlighted in Fig. 3. By multiplying the gradient by the input value, the method implicitly assumes that the relationship between the input and the model's output is locally linear near zero, which can produce misleading interpretations of feature contributions due to the inherent non-linearity of survival prediction curves. The cubic shape of the survival prediction curves leads to approximately parabolic relevance curves when considering time-independent feature effects, primarily because the survival probability tends to exhibit fewer changes at the extreme ends of time. This behavior is particularly pronounced in Cox-based NNs due to the shape assumptions inherent in their design. Since the survival curves in the DeepSurv model are constrained to be proportional, the resulting relevance curves also exhibit approximate proportionality. Consequently, in time-independent scenarios, CoxTime and DeepSurv yield similar results, indicating that CoxTime identifies the time-independence of the features.

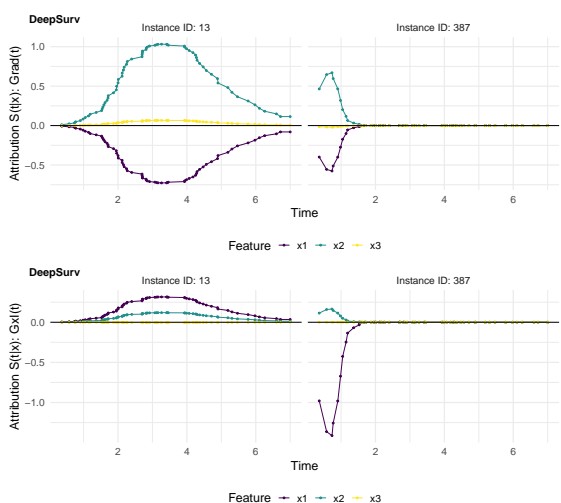

*Figure 3.* Grad(t) (top) and G×I(t) (bottom) relevance curves for selected observations using the DeepSurv model trained on the time-independent simulation dataset. The relevance values for each feature are represented by different colors (y-axis) and are plotted across time (x-axis), highlighting the temporal dynamics of feature contributions.

### 5.1.2. TIME-DEPENDENT EFFECTS

**Setup.** In this experiment, we generate synthetic data to demonstrate that gradient-based explanations can accurately capture variables with local time-dependent effects, provided the models correctly identify them. The dataset consists of $N = 10,000$ observations simulated from a Weibull model analogous to Sec. 5.1.1 with $\gamma = 1.5$. The features

include one time-dependent feature $x_1$ with a "harmful" effect for $t < 2$ and a "protective" effect for $t > 2$, as well as two time-independent features: one with a "harmful" effect ($x_2$) and one with a stronger "protective" effect ($x_3$). Additionally, there is one feature with no effect on the hazard ($x_4$). The maximum follow-up period is set to $t = 7$. Further details on the simulation settings, training process and fitted models are provided in Appendix A.2.

**Results.** These experiments are designed to demonstrate how the different gradient-based explanation methods capture the effects of time-dependent features. The relevance curves derived from output-sensitive methods (Figures A.12, A.14) effectively reveal the time-dependent effect of $x_1$ on the survival predictions, by indicating a positive effect at earlier times and a negative effect later on. This time-dependent effect is accurately captured by CoxTime and DeepHit (as illustrated in Fig. 4), but not by DeepSurv, which is inherently constrained by the PH assumption and thus unable to model time-dependence. These results underscore the ability of gradient-based methods to uncover such differences between models, focusing on explaining model behavior rather than data, and are thus valuable for assessing whether time-dependent variables are correctly modeled.

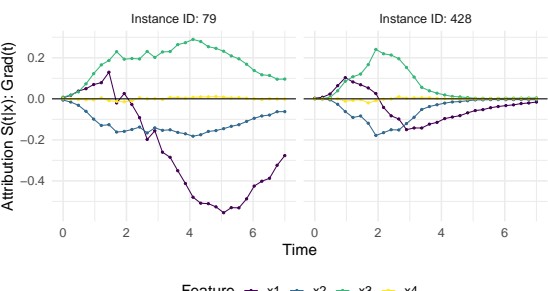

*Figure 4.* Grad(t) relevance curves for the selected observations and the DeepHit model trained on the time-dependent simulation dataset.

In addition to time-dependence in feature effects, difference-to-reference methods (i.e., IntGrad(t) and GradSHAP(t)) provide insights into the relative scale, direction, and magnitude of feature effects by comparing predictions to a meaningful reference, as displayed in Figures A.20, A.23 and A.26. *Contribution plots* (Figures A.21, A.24 and A.27) effectively visualize the normalized absolute contribution of each feature to the difference between reference and (survival) prediction over time, as shown for the GradSHAP(t) method and the CoxTime model for the two selected observations in Fig. 5. Complementarily, *force plots* (Figures A.22, A.25 and A.28) emphasize the relative contribution and direction of each feature at a set of representative survival times, likewise exemplarily highlighted in Fig. 5. For example, the opposite effects of low vs. high values

of $x_1$ are effectively captured in the plots. In observation 79, a low $x_1$ positively influences survival at later time points ($t > 2$) compared to the overall average survival in the dataset, resulting in its largest contributions occurring at these times. Conversely, in observation 428, a high $x_1$ induces substantial contributions at earlier time points ($t < 2$), but negatively impacts survival at later times, reflecting its early event as a consequence of the high $x_1$ and the strong negative effect of $x_3$. The average normalized absolute contribution, displayed on the right side of the contribution plots, offers a time-independent measure of feature importance, confirming the dominance of $x_3$ for the survival prediction of instance 428. Additionally, the visualizations suggest that CoxTime partially attributes the time-varying effect of $x_1$ to the other features, as the model, being non-parametric and lacking explicit knowledge of the time-dependent functional form, struggles to precisely disentangle and localize this effect.

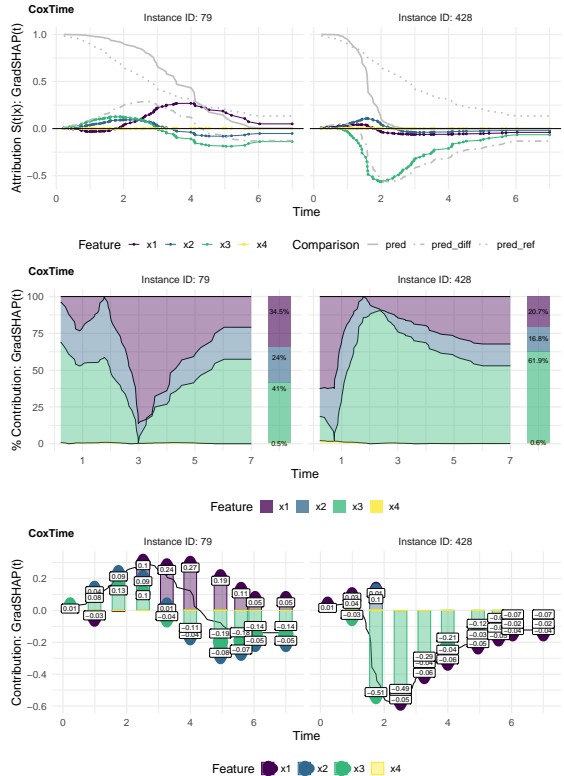

*Figure 5.* GradSHAP(t) relevance curves, with corresponding survival prediction curves, reference curve and their difference (top), contribution plots (middle) and force plots (bottom) for the selected observations and the CoxTime model trained on the time-dependent simulation dataset.

### 5.2. GradSHAP(t) vs SurvSHAP(t)

One of the most established XAI methods are SHAP values, which – rooted in game theory – offer intuitive interpretations by measuring the "gain" of each feature to a prediction (Chen et al., 2023). In the survival context, the only existing estimation approach is the model-agnostic SurvSHAP(t) method (Krzyziński et al., 2023). This sample-based strategy becomes computationally inefficient for high-dimensional feature spaces or deep NNs. Our proposed extension, GradSHAP(t), provides a model-specific counterpart, leveraging gradients for more efficient and scalable attribution. In the following, we compare both methods in simulated settings in terms of accuracy, runtime, and their ability to correctly estimate global feature rankings, particularly in comparison to SurvLIME. Further details of the simulations and comparable results for the other not-shown model classes can be found in Appendix A.3.

**Local accuracy.** To evaluate the accuracy of the method, we use the time-dependent adaption of the local accuracy measure proposed in Krzyziński et al. (2023), which is a function of $t$:

$$\sqrt{\frac{\mathbb{E}_{\boldsymbol{x}}\left[\left(f(t|\boldsymbol{x}) - \mathbb{E}_{\tilde{\boldsymbol{x}}}[f(t|\tilde{\boldsymbol{x}})] - \sum_{j=1}^{p} R_j(t|\boldsymbol{x})\right)^2\right]}{\mathbb{E}_{\boldsymbol{x}}[f(t|\boldsymbol{x})]}}. \quad (5)$$

This metric measures the decomposition error on a global level and normalizes it against the mean prediction at each time point. The simulation setup follows the same structure as described in Sec. 5.1.1, with $p = 30$ features and $1,000$ training instances. The features have a uniformly increasing effect strength from 0 to 1 on the log-hazard, with alternating signs. The results for a DeepSurv model are shown in Fig. 6. In our simulations, all 100 test samples are explained using the entire test set as the baseline dataset, while the number of interval samples (indicated in parentheses) for estimating the integral in GradSHAP(t) is varied. Our GradSHAP(t) explanations provide highly accurate approximations, even with a limited number of integration samples. Upon user demand, accuracy can be further improved by increasing the number of integration samples, albeit at the cost of a longer computation time.

**Runtime.** A key advantage of GradSHAP(t) is its superior scalability in higher dimensions, making it especially valuable for deep NNs. Fig. 6 illustrates the runtime of SurvSHAP(t) and GradSHAP(t) (with 5, 25, and 30 integration samples) as a function of the input dimension $p$ for DeepHit and DeepSurv. The plot demonstrates that the gradient-based method is significantly faster and maintains good scalability even for larger $p$. However, it also highlights the trade-off between accuracy and runtime: increasing the number of integration points results in longer yet still competitive computation times. This is mainly due to the computation of the gradients of a temporary instance

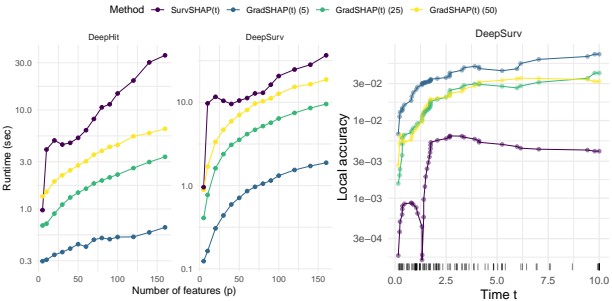

Figure 6. Runtime (left) and local accuracy (right) comparison of SurvSHAP(t) and GradSHAP(t) with varying numbers of integration samples (5, 25, 50) showing the trade-off between accuracy and efficiency.

for each sample and each integration point, leading to a computational complexity of $\mathcal{O}(n \cdot n_{\text{samples}} \cdot n_{\text{int}})$.

**Practical feasibility.** To complement our simulation-based runtime analysis, we demonstrate that GradSHAP(t) enables SHAP-like explanations for high-dimensional inputs much faster than its model-agnostic counterpart SurvSHAP(t) in a real-world example. Using the dataset and DeepHit model from Sec. 5.3, we replace the original ResNet34 (He et al., 2016) with a smaller variant (ResNet18) and reduce the input image size from 226x226 to 32x32, which still constitutes a high-dimensional input space. In this experiment, we explain a single instance and compare GradSHAP(t) and SurvSHAP(t) across several parameter settings in terms of runtime and time-averaged instance-wise local accuracy, i.e., an aggregated measure for the prediction-normalized decomposition goal of a single explanation (local explanation level)

$$\mathbb{E}_t \left[ \sqrt{\frac{\left( f(t|\boldsymbol{x}) - \mathbb{E}_{\tilde{\boldsymbol{x}}}[f(t|\tilde{\boldsymbol{x}})] - \sum_{j=1}^{p} R_j(t|\boldsymbol{x}) \right)^2}{\mathbb{E}_{\boldsymbol{x}}\left[ f(t|\boldsymbol{x}) \right]}} \right]. \quad (6)$$

Figure 7 summarizes the results. The left panel shows the time-averaged instance-wise local accuracy values (lower is better) across all tested configurations. Overall, Grad-SHAP(t) achieves similar or even slightly better accuracy levels to SurvSHAP(t). The right panel compares the runtime on a logarithmic scale. While both methods scale with the number of samples, GradSHAP(t) is substantially faster, particularly in settings with a higher number of samples (e.g., 50), requiring only a fraction of the computation time compared to SurvSHAP(t). For example, even in its most compute-intensive configuration, GradSHAP(t) completes in a couple of seconds, whereas SurvSHAP(t) requires several minutes for similar accuracy. This demonstrates that GradSHAP(t) offers a feasible trade-off between estimation

accuracy and efficiency with dramatically reduced runtime for deep survival models, which is an essential advantage for scaling interpretability in high-dimensional survival settings.

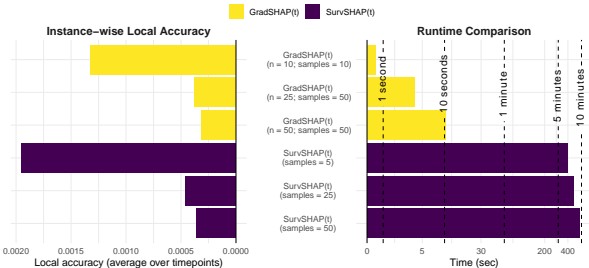

Figure 7. Instance-wise local accuracy (left) and runtime (right) for GradSHAP(t) and SurvSHAP(t) on a real-world survival model with high-dimensional image inputs. Both methods achieve similar accuracy, while GradSHAP(t) is considerably faster in all settings.

**Global importance ranking.** Beyond patient-wise local effects, it is important to assess whether features consistently rank as influential on a global level. Stable importance rankings indicate that the model captures robust patterns rather than instance-specific artifacts. For this simulation, we use $p = 5$ features, each having an evenly increasing effect on the log-hazard function with alternating signs. Additionally, we include the importance ranking of SurvLIME weights as a competing global XAI method. SurvLIME is an extension of the LIME framework adapted for survival models, fitting local surrogates. As shown in Fig. 8, the feature importance rankings of SurvSHAP(t) and GradSHAP(t) are nearly identical and agree with the data-generating process. The discrepancy to SurvLIME is consistent with observations reported in previous studies (Krzyziński et al., 2023).

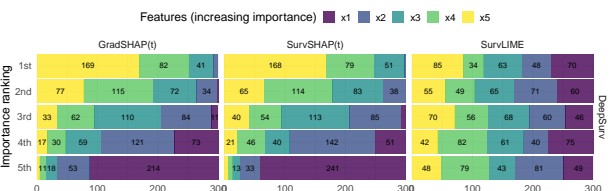

Figure 8. Global importance rankings for GradSHAP(t), SurvSHAP(t), and SurvLIME across 300 test samples. Grad-SHAP(t) and SurvSHAP(t) show consistent rankings aligned with the data-generating process.

### 5.3. Example on Real Multi-modal Medical Data

Despite the critical role of time-to-event prediction in medical decision-making, deep learning remains underutilized in this domain, even with its success in other medical ap-

plications. Our primary motivation for developing gradient-based explanation methods for survival deep learning is to help researchers harness the unique abilities of deep learning to extract and integrate complex features from high-dimensional, unstructured data while ensuring interpretability, transparency in individual-level decision-making, and facilitating new domain knowledge discovery.

In order to show a use case, we apply the methods to a CNN-based extension of a DeepHit model trained on a real-world multi-modal medical dataset predicting overall survival in diffuse gliomas (Mobadersany et al., 2018). We use four tabular features selected based on previous results (age, sex, absence or presence of IDH mutation and 1p/19q codeletion) and histologic images of the regions of interest of whole-slide image tissue sections from formalin-fixed, paraffin-embedded specimens from The Cancer Genome Atlas (TCGA) Lower-Grade Glioma (LGG) and Glioblastoma (GBM) projects. The molecular features are known to be helpful for predicting survival in gliomas and other brain tumors. Based on the WHO's histologic classification of gliomas (Park et al., 2023), the isocitrate dehydrogenase mutation (IDH) involves alterations in the IDH1 or IDH2 genes and is associated with a more favorable prognosis. The 1p/19q codeletion refers to the simultaneous loss of parts of chromosomes 1 and 19. The absence of a 1p/19q codeletion is associated with more invasive and treatment-resistant gliomas, leading to a worse survival prognosis. For the image data of resized shape 226x226, we use a standard ResNet34 architecture (He et al., 2016). The high-level representations extracted from the ResNet are flattened to 256 features and fused with the tabular data. This combined representation is then passed into a final dense network, which serves as the base model for a multi-modal DeepHit architecture. This multi-modal model is trained on a total of $1,239$ training samples and evaluated on 266 test samples, achieving a C-index of $0.713$ and an integrated Brier score of $0.092$. We use the standard DeepHit loss function with an $\alpha$-value of $0.5$ to balance the rank loss and log-likelihood loss equally. For an individual-level temporal explanation, we use GradSHAP(t) with 100 baseline samples and 20 integration points, computation took almost 12 minutes.[5]

Fig. 9 shows the force plot of GradSHAP(t) explanation for a 43-year-old male patient with an IDH mutation but no 1p/19q codeletion. The black solid line represents the difference between the patient's survival prediction and the dataset-wide average. Initially, the line remains above the x-axis, suggesting a survival advantage in the early months – likely due to the protective effect of the IDH mutation, which contributes positively at all time points. However, the curve

---

[5]Attempts to apply SurvSHAP(t) with the same setup but just a single baseline sample did not complete within a week and used almost 800GB RAM, showcasing the advantage of gradient-based explanations for deep survival models.

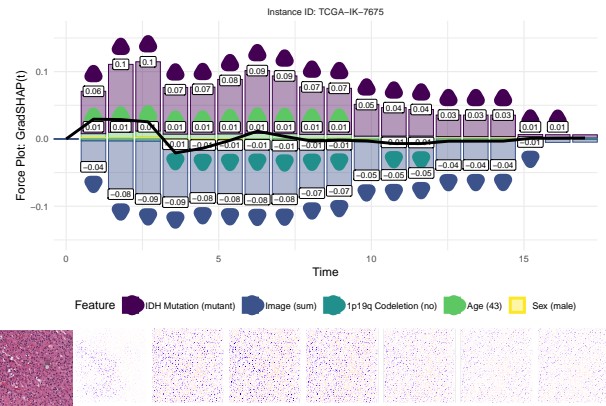

*Figure 9.* GradSHAP(t) explanation for a multi-modal DeepHit model, showing temporal contributions of tabular genetic and clinical features (top). Below, corresponding image patches over time are visualized, with the original histology image shown at the bottom left.

eventually shifts below the x-axis, indicating a survival disadvantage over time, potentially driven by the absence of the 1p/19q codeletion and the diminishing effect of the IDH mutation at later time points. The aggregated effect of the image modality consistently indicates a negative impact on the survival probability. This is further evidenced in the image explanations (bottom), predominantly highlighting cells with a negative contribution.

## 6. Conclusion

In this work, we introduce a set of novel model-specific local XAI methods for survival deep learning. The methods are the first to provide time-dependent explanations for multi-modal data, including images, in the survival setting, but is likewise easily generalizable to any functional outputs. Our work equips practitioners with a toolkit to derive meaningful insights from fitted survival NNs while accounting for underlying model assumptions grounded in survival analysis theory and the interpretability offered by different gradient-based techniques. This promotes transparency, accountability, and fairness in sensitive applications such as clinical decision-making, the development of targeted therapies, medical interventions, and other healthcare contexts. However, it is important to note that these explanations do not imply causal relationships, as the models lack knowledge of the true causal structure of the data-generating process. Future work will extend these methods beyond right-censored survival to include competing risks, multi-state models, and recurrent events. We also aim to develop targeted XAI methods for deep learning that can detect both individual feature effects and interactions.

## Acknowledgements

The results shown here are in part based upon data generated by the TCGA Research Network: https://www.cancer.gov/tcga. This project and the authors were supported by the German Research Foundation (DFG), Grant Numbers: 437611051, 459360854.

## Impact Statement

This paper introduces a framework for interpreting individual predictions of survival analysis deep learning models, which has the potential to enable more transparent and actionable predictions of time-to-event outcomes in critical fields such as healthcare, insurance, and personalized medicine. Therefore, this work addresses ethical concerns regarding the opacity of deep learning models, ensuring that predictions can be understood and trusted by stakeholders. This transparency is crucial for mitigating biases, fostering equitable decision-making, and ensuring compliance with regulatory standards. It needs to be stressed, that interpretable machine learning methods are useful to discover knowledge, to debug or justify, as well as control and improve models and their predictions, but not to draw causal conclusions from the data. While there are risks of misuse and misunderstanding, we believe the net positive impact substantially outweighs the risks. Future societal consequences include enhanced patient care through personalized treatment plans, improved risk assessment in critical industries, and broader public trust in AI-driven systems.

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

# A. Experiments on Simulated Data

All simulations and real-data examples presented in this manuscript, along with the corresponding code, are available in our GitHub repository at https://github.com/bips-hb/Survival-XAI-ICML/ to ensure transparency and reproducibility.

## A.1. Time-independent Effects

For a chosen observation, the hazard function from which the data are generated is of the form:

$$h(t|\boldsymbol{x}) = \lambda\,\gamma\,t^{\gamma-1}\exp(1.7x_1 - 2.4x_2) \tag{7}$$

with $\lambda = 0.1$ and $\gamma = 2.5$ and $x_1, x_2, x_3 \sim \mathcal{N}(0,1)$. To generate the event times $t^{(i)}$ for instance $i$, the method of (Bender et al., 2005) is applied and the simsurv package (Brilleman et al., 2020) is used. Observations are artificially censored for $t^{(i)} \geq 7$.

The data is split into training ($9,500$ observations) and test set ($500$ observations) and DeepSurv, CoxTime and DeepHit models with two dense layers with 32 nodes are fit to the training data without tuning, using 500 epochs, early stopping, a batch size of $1,024$ and a dropout probability of $0.1$ applied to all layers. For any other hyperparameters, including the activations the default values set in the pycox (Kvamme et al., 2019) Python package are used, which are based on the default values suggested by (Katzman et al., 2018; Kvamme et al., 2019; Lee et al., 2018). More details are provided in our code supplement.

The models' performance expressed in the Brier score is shown in Fig. A.1 and Table A.1 shows the Concordance index (C-index) and the Integrated Brier Score (IBS) as aggregated performance measures. DeepSurv slightly outperforms CoxTime and DeepHit in the Brier Score, likely because of its inherent assumption of proportional hazards, which the data simulated from a Cox model with time-independent effects are subject to. Even though CoxTime performs similarly to DeepSurv in Brier Score, this does not necessarily imply conformity to the PH assumption; it has to be assessed, for instance using gradient-based explanations. DeepHit has a higher C-index and IBS, suggesting poorly calibrated probabilistic predictions compared to DeepSurv and CoxTime, which is a consequence of the emphasis on C-index maximization in the loss function.

*Table A.1.* Performance metrics for different survival models fitted on time-independent simulation data (C-index: higher is better, 0.5 indicates random prediction; IBS: lower is better).

| MODEL | C-INDEX ($\uparrow$) | IBS ($\downarrow$) |
|---|---|---|
| COXTIME | 0.807 | 0.1 |
| DEEPSURV | 0.809 | 0.099 |
| DEEPHIT | 0.809 | 0.142 |

Two observations are randomly chosen to illustrate the gradient-based explanation methods for survival deep learning models delineated in Sec. 4. Their respective feature values and observed survival times are denoted in Table A.2. The survival curves predicted by the selected survival NN models are shown in Fig. A.9.

The relevance values for each feature are represented by different colors are plotted across time in Figures A.2-A.8 for the selected observations and models, highlighting the temporal dynamics of feature contributions.

*Table A.2.* Feature values, observed survival time and event status of two randomly chosen observations (ID 13 and ID 387) from the test set of the simulated dataset with time-independent feature effects

| ID | TIME | STATUS | X1 | X2 | X3 |
|---|---|---|---|---|---|
| 13 | 2.665 | 1 | -0.435 | 0.1162 | -0.081 |
| 387 | 0.958 | 1 | 2.455 | 0.2462 | -0.043 |

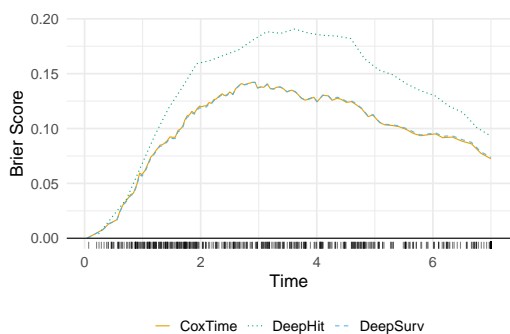

*Figure A.1.* Performance of DeepSurv, CoxTime, and DeepHit models over time measured by Brier score (lower is better; Brier score of 0.25 indicates prediction at random).

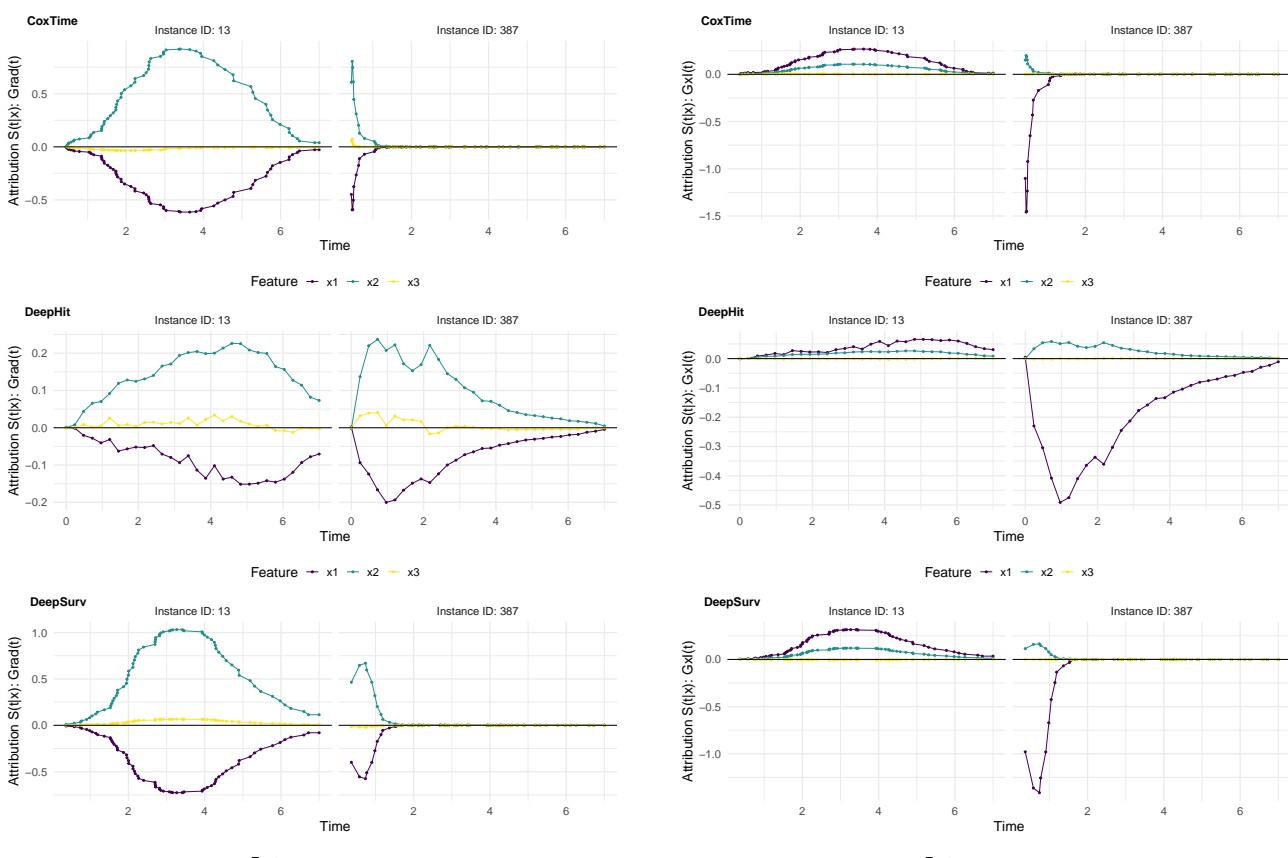

*Figure A.2.* Grad(t) relevance curves for the selected observations and models trained on the time-independent simulation dataset. The relevance values for each feature are represented by different colors (y-axis) and are plotted across time (x-axis).

*Figure A.3.* G×I(t) relevance curves for the selected observations and models trained on the time-independent simulation dataset. The relevance values for each feature are represented by different colors (y-axis) and are plotted across time (x-axis).

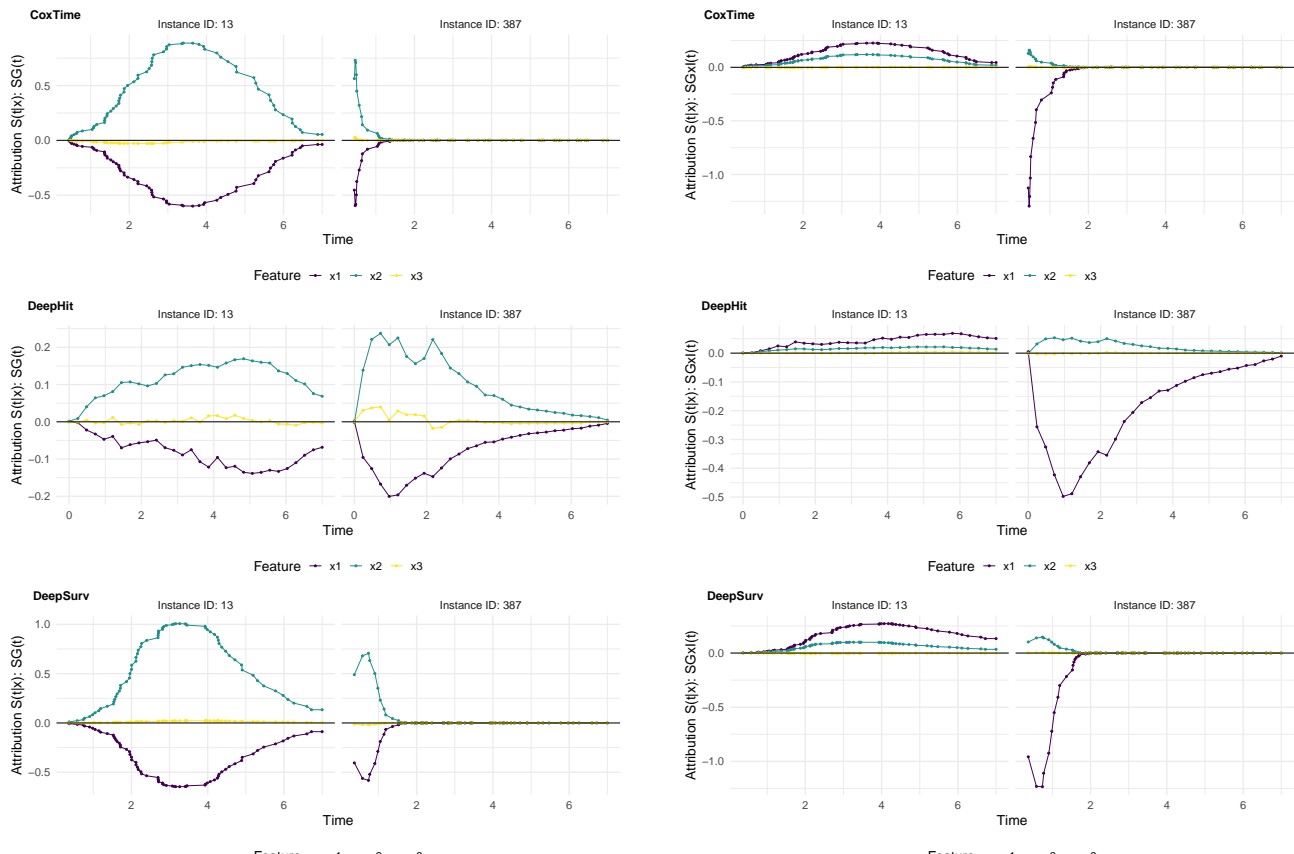

*Figure A.4.* SG(t) relevance curves for the selected observations and models trained on the time-independent simulation dataset. The relevance values for each feature are represented by different colors (y-axis) and are plotted across time (x-axis).

*Figure A.5.* SG×I(t) relevance curves for the selected observations and models trained on the time-independent simulation dataset. The relevance values for each feature are represented by different colors (y-axis) and are plotted across time (x-axis).

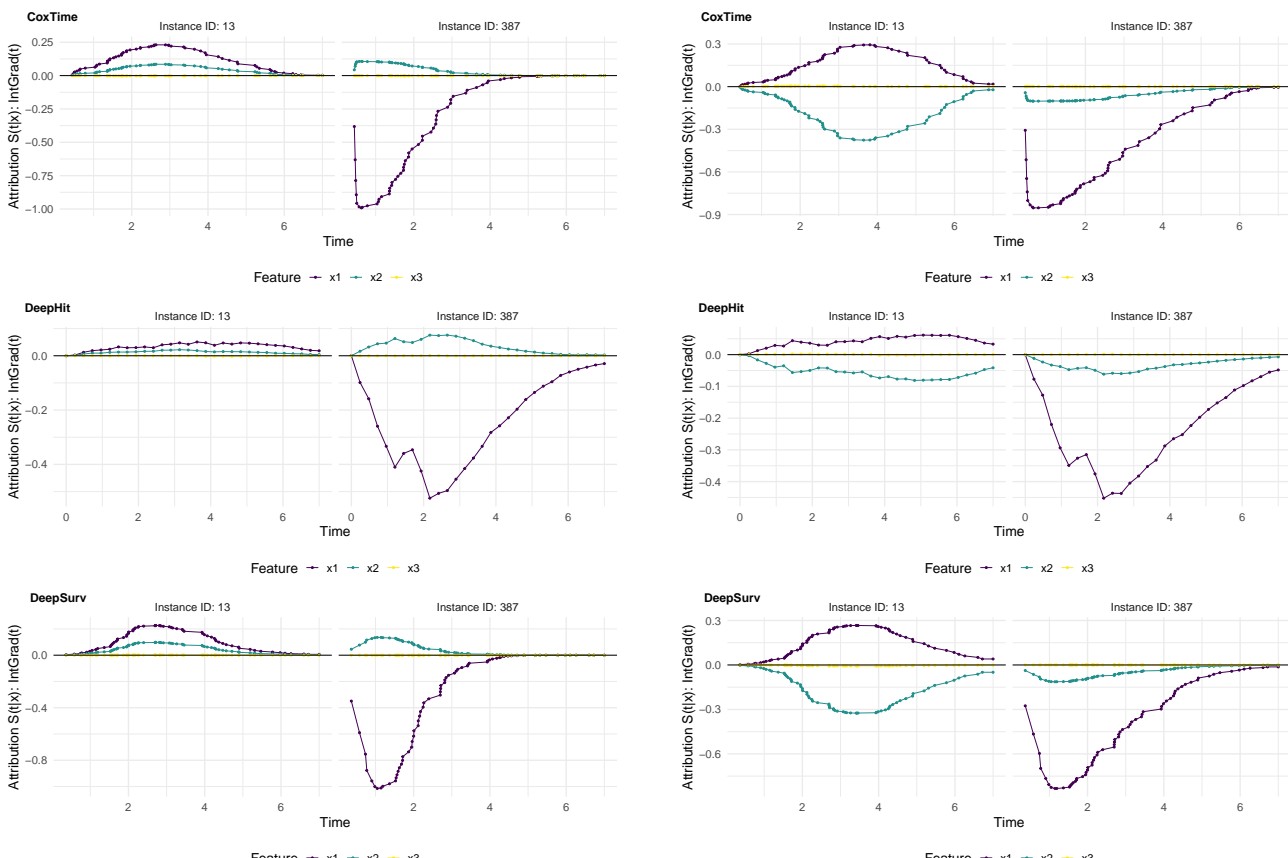

Figure A.6. IntGrad(t) relevance curves for the selected observations and models trained on the time-independent simulation dataset. The reference value is the null observation (all feature values set to zero). The relevance values for each feature are represented by different colors (y-axis) and are plotted across time (x-axis).

Figure A.7. IntGrad(t) relevance curves for the selected observations and models trained on the time-independent simulation dataset. The reference value is the mean observation (feature values set to the average over all observations). The relevance values for each feature are represented by different colors (y-axis) and are plotted across time (x-axis).

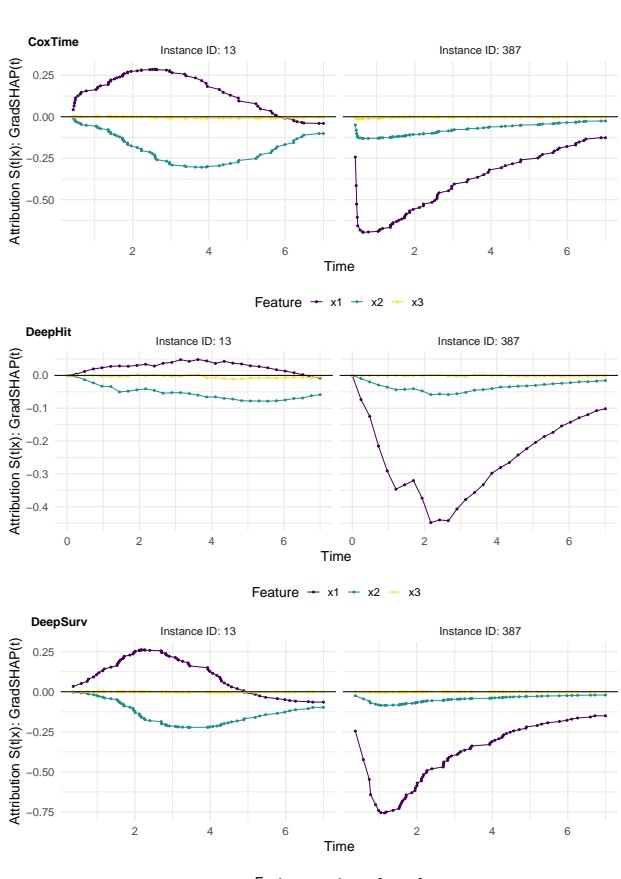

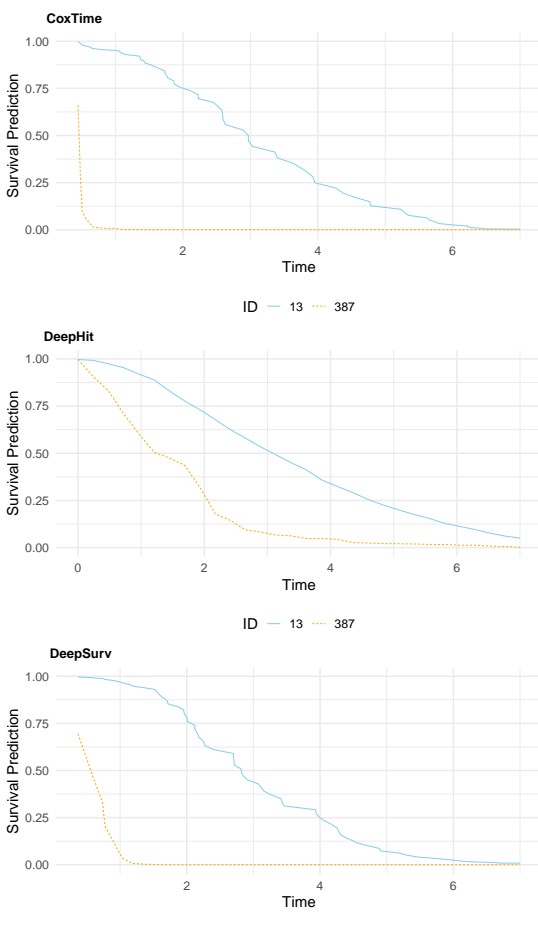

*Figure A.8.* GradSHAP(t) relevance curves for the selected observations and models trained on the time-independent simulation dataset. The relevance values for each feature are represented by different colors (y-axis) and are plotted across time (x-axis).

*Figure A.9.* Predicted survival curves of CoxTime, DeepHit and DeepSurv model for randomly chosen observations (blue: ID 13, orange: ID 387). The models unanimously predict a higher probability of survival at any given time point for ID 13.

## A.2. Time-dependent Effects

For a chosen observation, the hazard function from which the data are generated is of the form:

$$h(t|\boldsymbol{x}) = \lambda \gamma \, t^{\gamma-1} \exp(-3x_1 + 1.7x_2 - 2.4x_3 + 6x_1 \log(t)) \tag{8}$$

with $\lambda = 0.1$ and $\gamma = 1.5$ and $x_1 \sim \mathcal{U}(0,1)$, $x_2, x_3 \sim \mathcal{N}(0,1)$, $x_4 \sim \mathcal{U}(-1,1)$. To generate the event times $t^{(i)}$ for instance $i$, the method of (Bender et al., 2005) is applied and the `simsurv` package (Brilleman et al., 2020) is used. Observations are artificially censored for $t^{(i)} \geq 7$. The Kaplan-Meier survival curves grouped by low and high values of feature $x_1$ demonstrate the time-dependent nature of its effect. Individuals with high values of $x_1$ initially show a higher average probability of survival at earlier time points ($t < 2$), but their survival probability declines more rapidly over time, leading to a lower predicted probability of survival at later time points ($t > 2$). A real-world example of such an effect could be cancer medication that provides strong early benefits by effectively slowing tumor progression or reducing symptoms. However, over time, the medication's efficacy might diminish due to drug resistance or cumulative side effects, resulting in worse long-term outcomes for patients compared to those on lower dosage treatments.

*Table A.3.* Performance metrics for different survival models fitted on time-dependent simulation data (C-index: higher is better, 0.5 indicates random prediction; IBS: lower is better).

| MODEL | C-INDEX ($\uparrow$) | IBS ($\downarrow$) |
|---|---|---|
| COXTIME | 0.85 | 0.058 |
| DEEPSURV | 0.86 | 0.06 |
| DEEPHIT | 0.805 | 0.095 |

The data is split into training ($9,500$ observations) and test set ($500$ observations) and DeepSurv, CoxTime and DeepHit models with two dense layers of 32 hidden nodes are fit to the training data without tuning, using 500 epochs, early stopping, a batch size of $1,024$ and a dropout probability of $0.1$ applied to all layers. For any other hyperparameters, including the activations, the default values set in the `pycox` (Kvamme et al., 2019) Python package are used, which are based on the default values suggested by (Katzman et al., 2018; Kvamme et al., 2019; Lee et al., 2018). More details are provided in our code supplement.

The models' performance expressed in the Brier score is shown in Fig. A.10 and Table A.3 shows the Concordance index (C-index) and the Integrated Brier Score (IBS) as aggregated performance measures. CoxTime slightly outperforms DeepSurv and DeepHit in the Brier Score, likely because it is able to appropriately capture the violation of the assumption of proportional hazards in the time-dependent effects simulation. However, a superior performance does not necessarily imply that feature effects are captured correctly; this has to be assessed, for instance, using gradient-based explanations. DeepHit has a lower C-index and higher IBS, suggesting poorly calibrated probabilistic predictions as well as poor discriminatory power compared to DeepSurv and CoxTime, perhaps as a result of the Cox-based nature of the simulation.

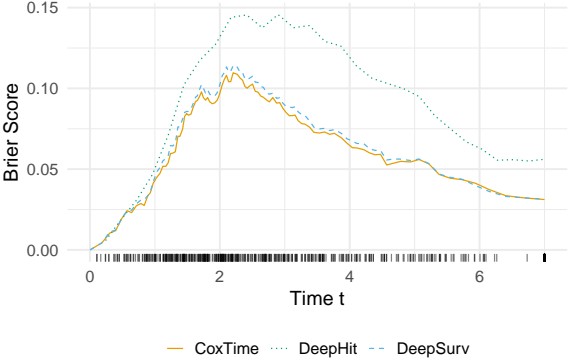

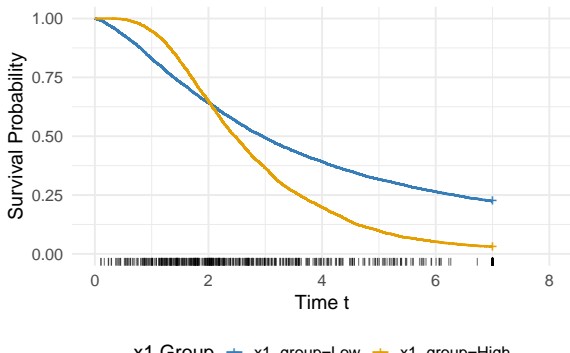

*Figure A.10.* Performance of DeepSurv, CoxTime, and DeepHit models over time measured by Brier score (lower is better; Brier score of 0.25 indicates prediction at random)

*Figure A.11.* Kaplan-Meier survival curves comparing individuals based on high (orange) and low (blue) values of feature $x_1$. Each curve represents the estimated survival rate for the corresponding group.

*Table A.4.* Feature values, observed survival time and event status of two randomly chosen observations (ID 79 and ID 428) from the test set of the simulated dataset with time-dependent feature effects.

| ID | TIME | STATUS | x1 | x2 | x3 | x4 |
|----|------|--------|-------|--------|--------|--------|
| 79 | 4.306 | 1 | 0.194 | -0.307 | -0.148 | 0.637 |
| 428 | 1.417 | 1 | 0.753 | -0.378 | -1.15 | -0.642 |

Two observations are randomly chosen to illustrate the gradient-based explanation methods for survival deep learning models delineated in Sec. 4. Their respective feature values and observed survival times are denoted in Table A.4. The survival curves predicted by the selected survival NN models are shown in Fig. A.19.

We propose several approaches to effectively visualize relevance values while incorporating the temporal dimension for difference-to-reference methods, beyond merely plotting the computed relevance values for individual features over time. In difference-to-reference methods, relevance values explain the deviation between the prediction (i.e., the predicted survival curve) for a selected observation and a chosen reference curve (e.g., the predicted survival curve for an observation where all feature values are set to zero, or where feature values are set to their respective means). To enhance clarity, we can plot the prediction (`pred`), the reference prediction (`pred_ref`), and their difference (`pred_diff`) alongside the relevance curves, as demonstrated in Figures A.20, A.23, and A.26. Contribution plots (Figures A.21, A.24, A.27) visualize the absolute, normalized feature-wise contributions to the prediction to reference difference, with the contributions represented as shaded areas colored by feature. These plots highlight how each feature relevances the prediction-to-reference difference. The absolute normalized contributions are calculated as

$$R_j^{t,\text{norm}} = \frac{|R_j^t|}{\sum_k |R_k^t|}, \tag{9}$$

and are then plotted in a stacked form, maintaining the feature order by using the cumulative sum across the features. This provides insight into the magnitude of each feature's percentage contribution to the prediction-to-reference difference at each point in time. Furthermore, the absolute normalized contributions can be averaged over time to derive a time-independent local feature importance measure, which highlights the overall impact of each feature across the entire time period:

$$R_j^{\text{norm}} = \frac{1}{|t|} \sum_t \left( \frac{|R_j^t|}{\sum_k |R_k^t|} \right), \tag{10}$$

which is plotted on the right hand side of Figures A.21, A.24, A.27. The force plots in Figures A.22, A.25, and A.28 are the time-dependent equivalent of the well-established SHAP force plots, illustrating how individual feature contributions combine to explain the prediction-to-reference difference. These plots provide a detailed breakdown of the magnitude and direction of the factors influencing the prediction-to-reference difference. A representative set of equidistant observed survival time points (e.g., 10 points) is selected, and the contribution of each feature is visualized using stacked bar plots. The direction of each feature's effect is emphasized by upturned arrows for positive contributions and downturned arrows for negative contributions. Different colors represent the respective features, and the magnitude of each feature's contribution is displayed as a label within its corresponding bar. The overall prediction-to-reference difference is shown as a black line for reference.

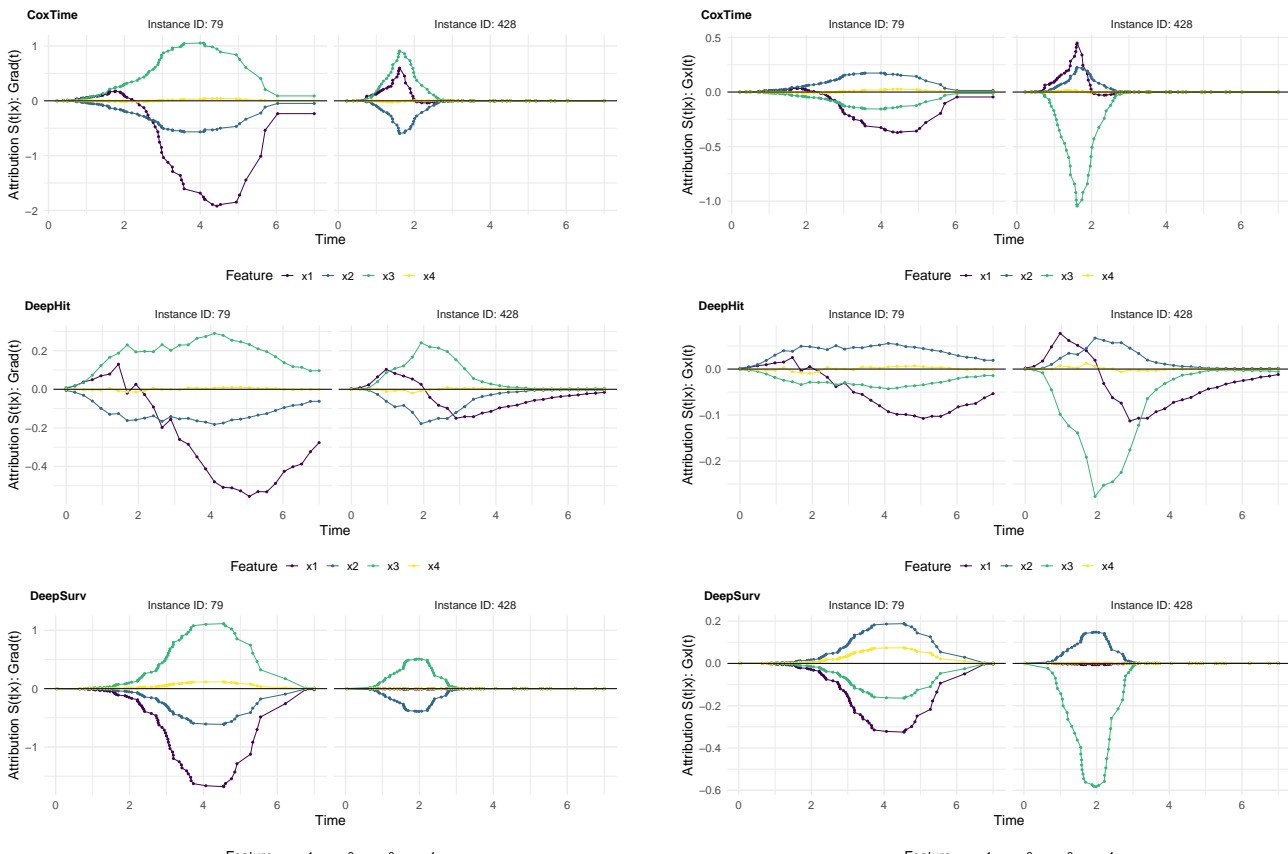

*Figure A.12.* Grad(t) relevance curves for the selected observations and models trained on the time-dependent simulation dataset. The relevance values for each feature are represented by different colors (y-axis) and are plotted across time (x-axis).

*Figure A.13.* G×I(t) relevance curves for the selected observations and models trained on the time-dependent simulation dataset. The relevance values for each feature are represented by different colors (y-axis) and are plotted across time (x-axis).

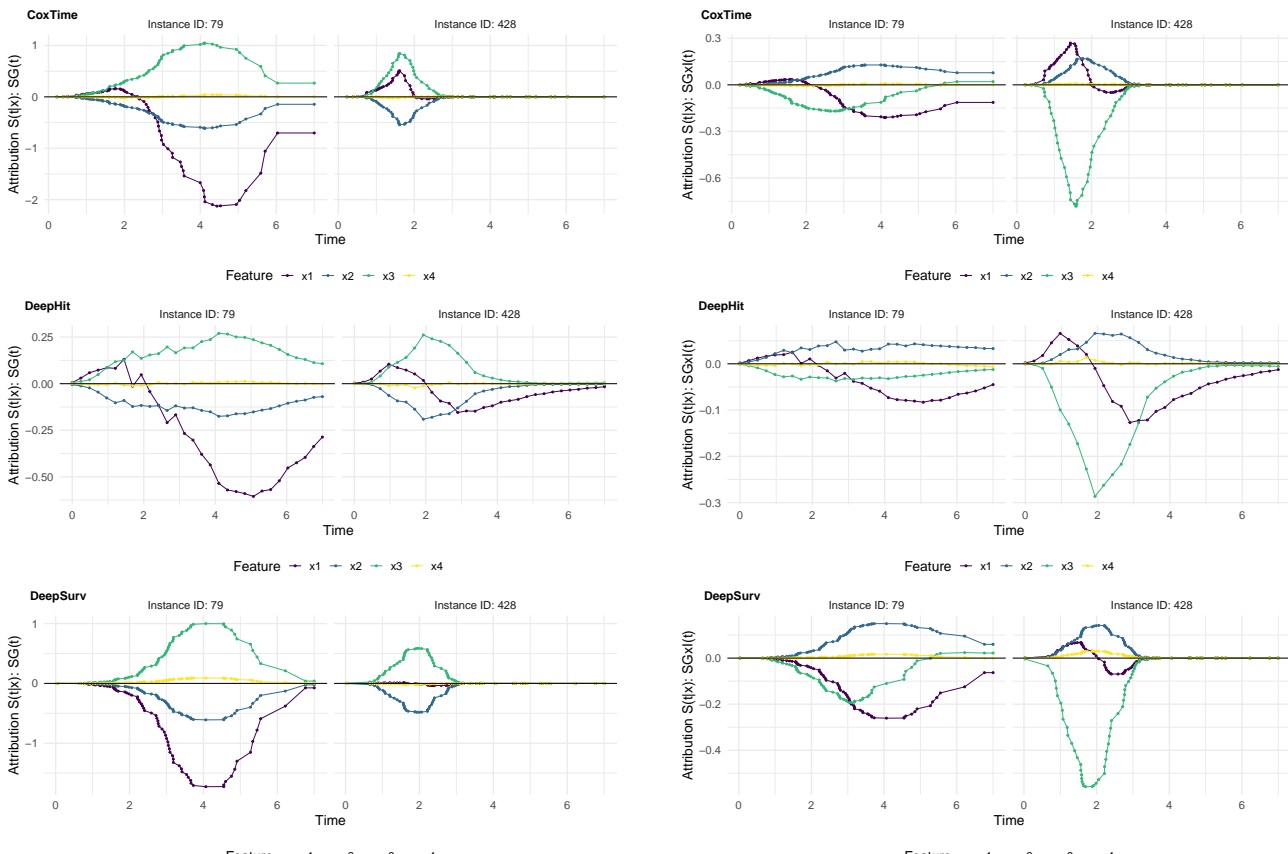

*Figure A.14.* SG(t) relevance curves for the selected observations and models trained on the time-dependent simulation dataset. The relevance values for each feature are represented by different colors (y-axis) and are plotted across time (x-axis).

*Figure A.15.* SG×I(t) relevance curves for the selected observations and models trained on the time-dependent simulation dataset. The relevance values for each feature are represented by different colors (y-axis) and are plotted across time (x-axis).

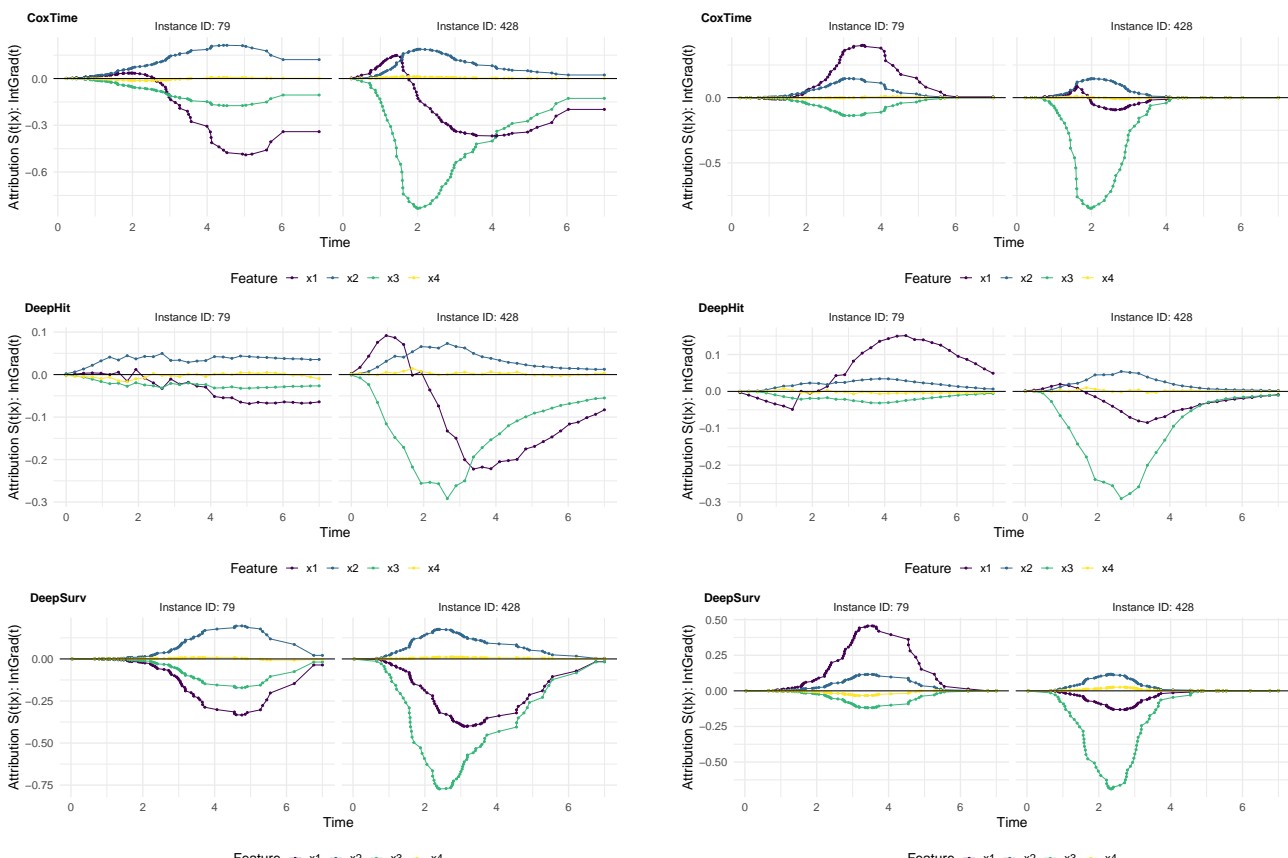

*Figure A.16.* IntGrad(t) relevance curves for the selected observations and models trained on the time-dependent simulation dataset. The reference value is the null observation (all feature values set to zero). The relevance values for each feature are represented by different colors (y-axis) and are plotted across time (x-axis).

*Figure A.17.* IntGrad(t) relevance curves for the selected observations and models trained on the time-dependent simulation dataset. The reference value is the mean observation (feature values set to the average over all observations). The relevance values for each feature are represented by different colors (y-axis) and are plotted across time (x-axis).

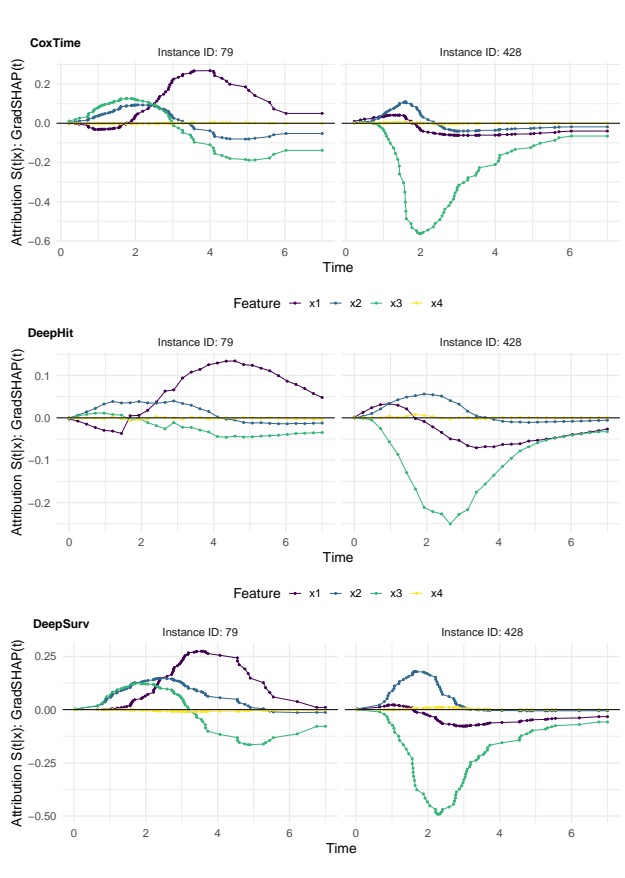

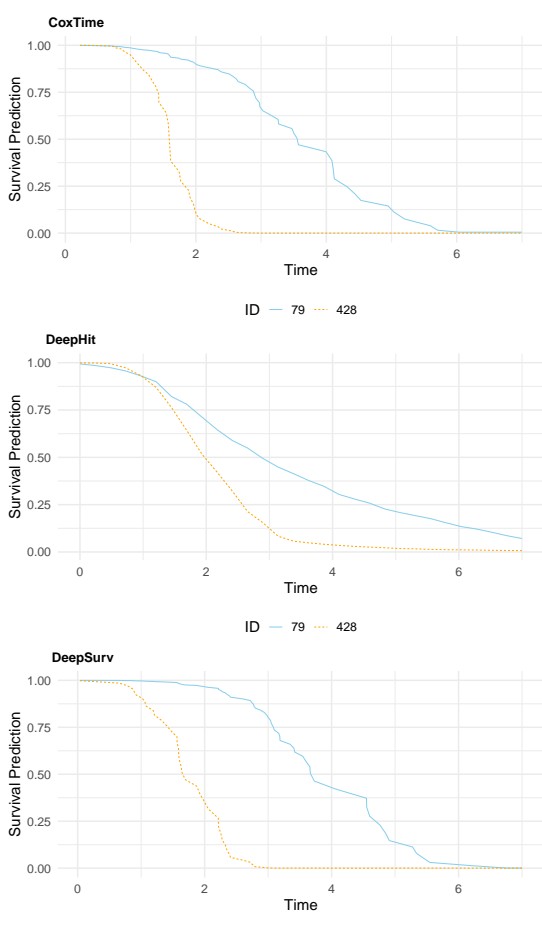

*Figure A.18.* GradSHAP(t) relevance curves for the selected observations and models trained on the time-dependent simulation dataset. The reference value is the mean observation (feature values set to the average over all observations). The relevance values for each feature are represented by different colors (y-axis) and are plotted across time (x-axis).

*Figure A.19.* Predicted survival curves of CoxTime, Deep-Hit and DeepSurv model for randomly chosen observations (blue: ID 79, orange: ID 428). The models unanimously predict a higher probability of survival at any given time point for ID 428.

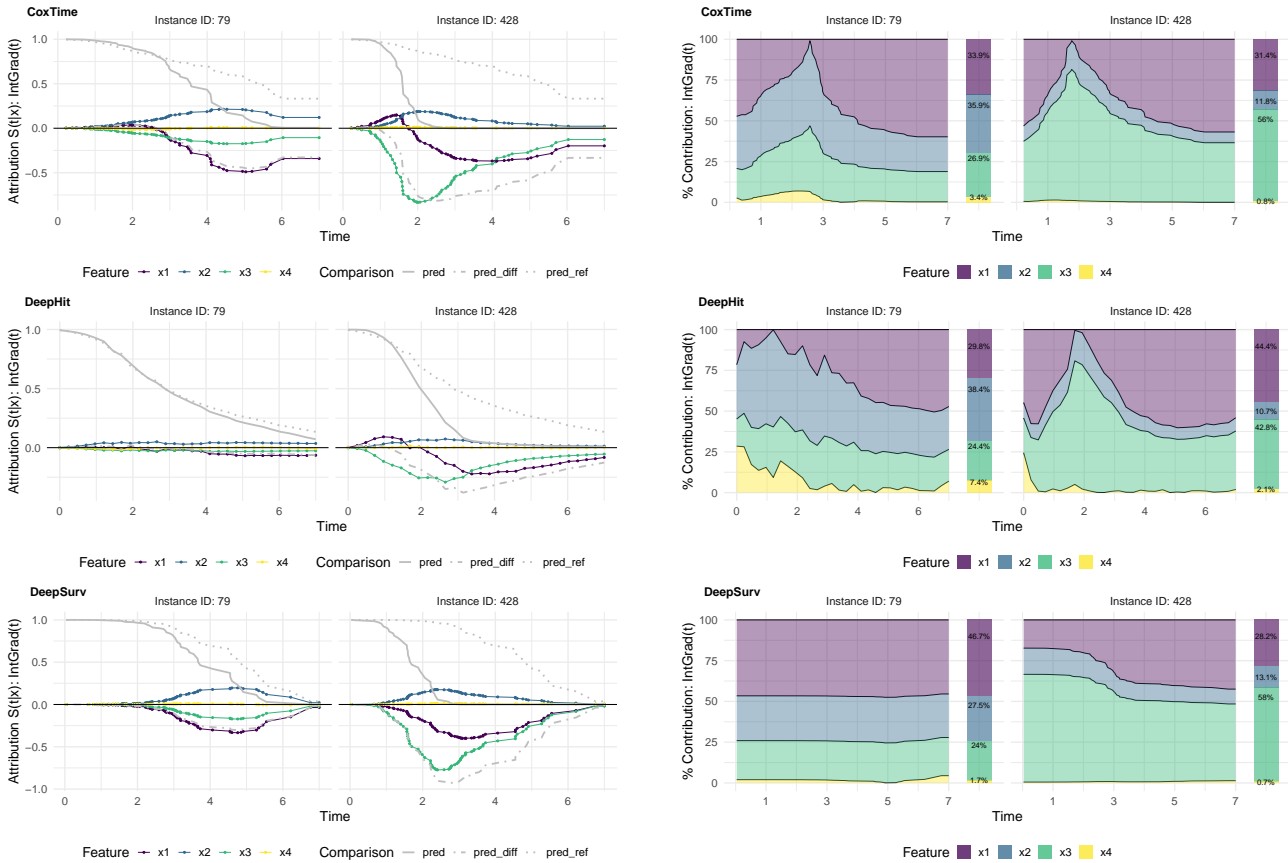

Figure A.20. IntGrad(t) relevance curves (yellow, turquoise, blue, purple) and predicted survival curves for the selected observations (ref), predicted survival curve for the reference observation (pred_ref) and their difference (pred_diff) for models trained on the time-dependent simulation dataset. The reference value is the null observation (all feature values set to zero).

Figure A.21. IntGrad(t) contribution plots for the selected observations and models trained on the time-dependent simulation dataset. The reference value is the null observation (all feature values set to zero).

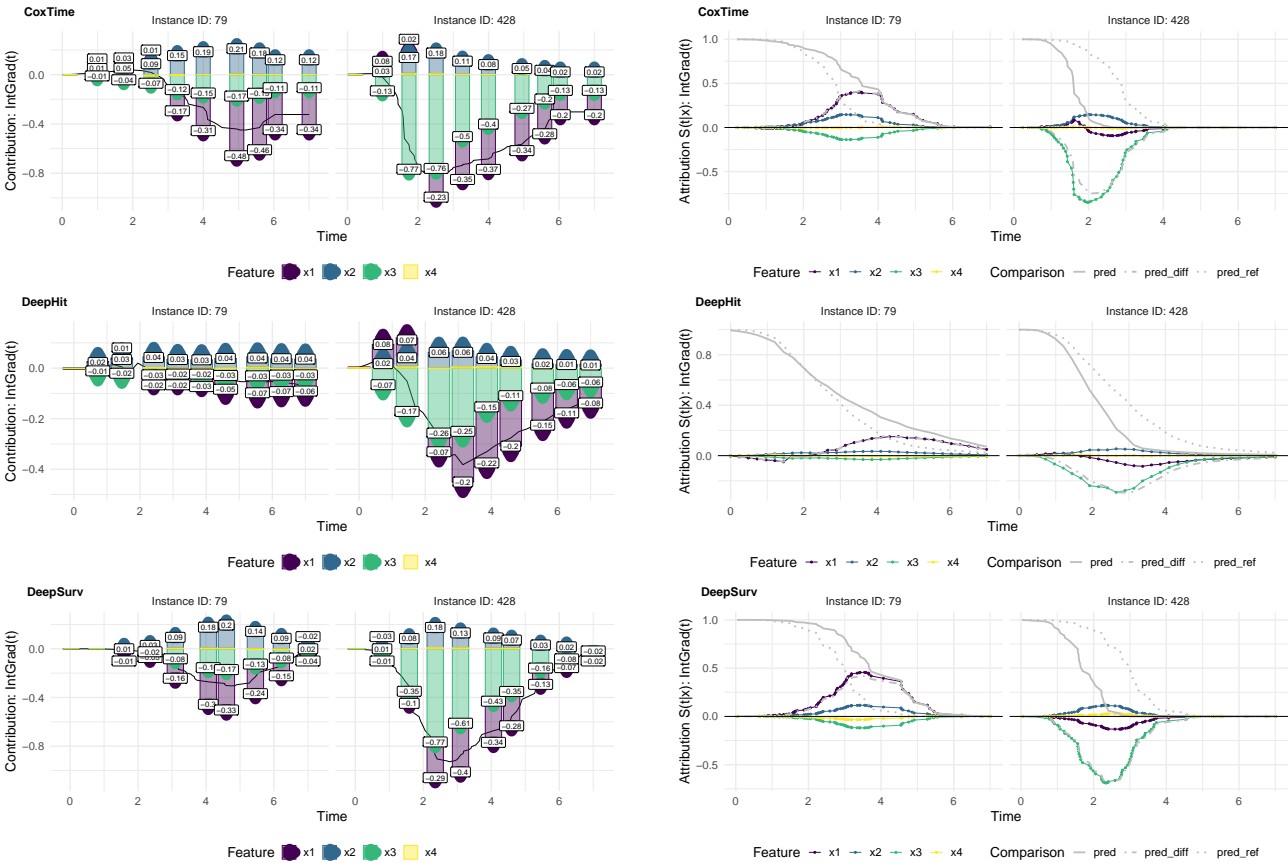

*Figure A.22.* IntGrad(t) force plots for the selected observations and models trained on the time-dependent simulation dataset. The reference value is the null observation (all feature values set to zero).

*Figure A.23.* IntGrad(t) relevance curves (yellow, turquoise, blue, purple) and predicted survival curves for the selected observations (ref), predicted survival curve for the reference observation (pred_ref) and their difference (pred_diff) for models trained on the time-dependent simulation dataset. The reference value is the null observation (feature values set to zero).

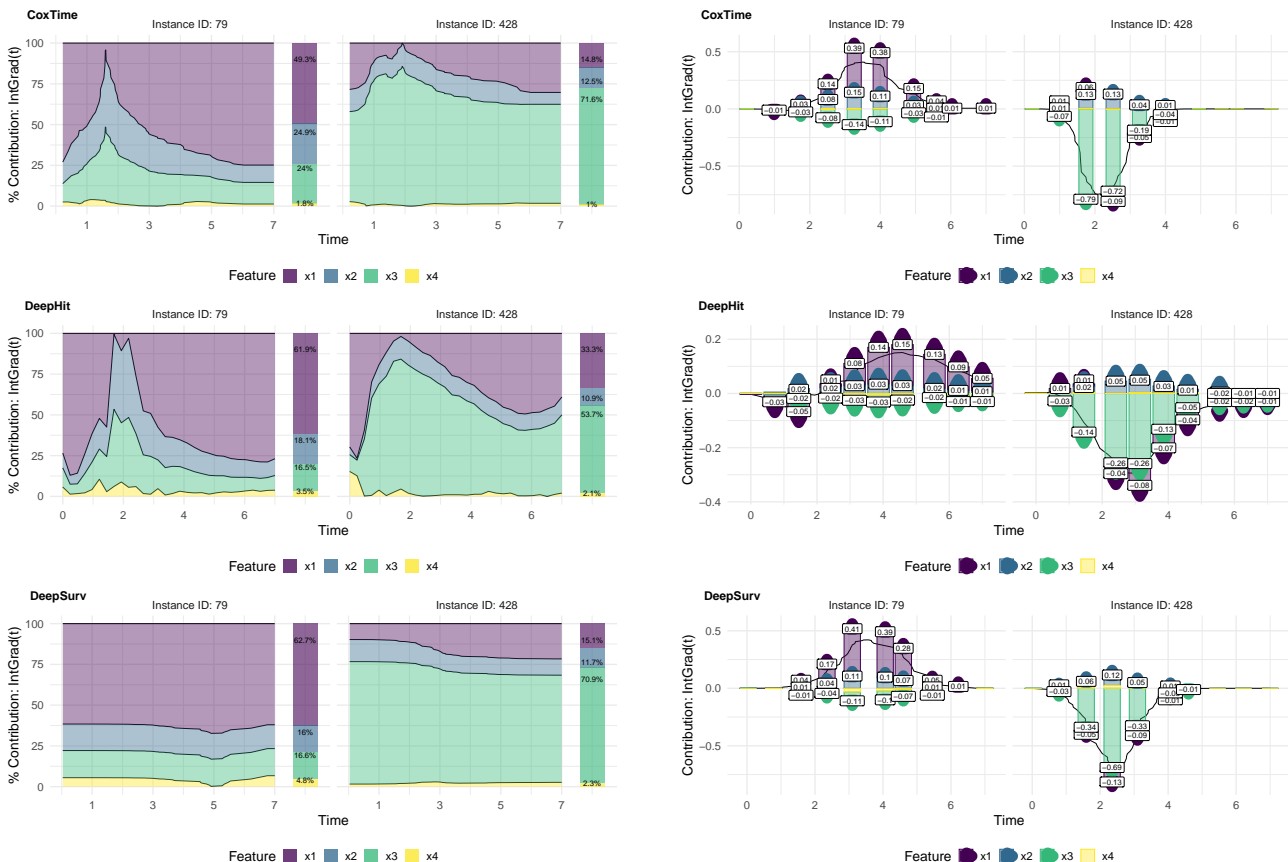

*Figure A.24.* IntGrad(t) contribution plots for the selected observations and models trained on the time-dependent simulation dataset. The reference value is the mean observation (feature values set to the average over all observations).

*Figure A.25.* IntGrad(t) force plots for the selected observations and models trained on the time-dependent simulation dataset. The reference value is the mean observation (feature values set to the average over all observations).

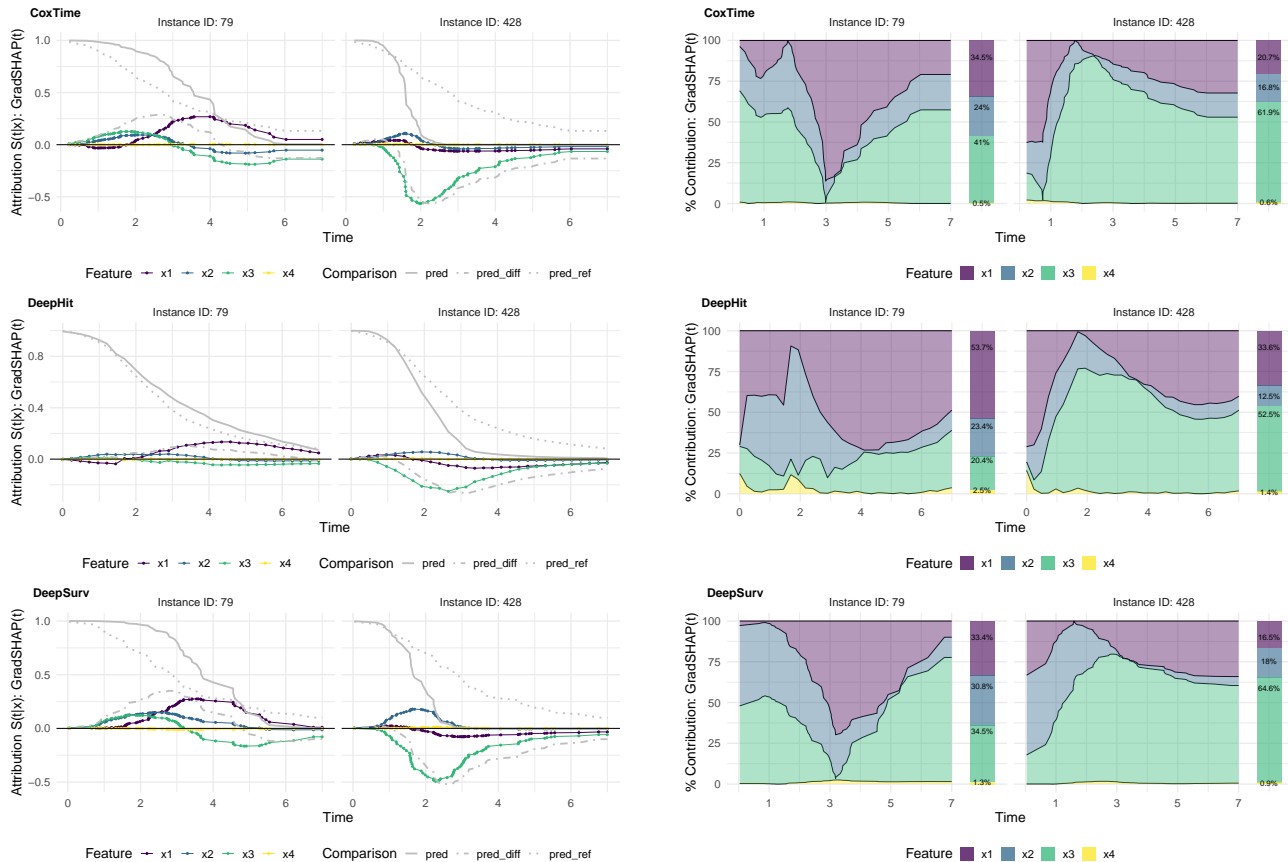

*Figure A.26.* GradSHAP(t) relevance curves (yellow, turquoise, blue, purple) and predicted survival curves for the selected observations (ref), predicted survival curve for the reference observation (pred_ref) and their difference (pred_diff) for models trained on the time-dependent simulation dataset.

*Figure A.27.* GradSHAP(t) contribution plots for the selected observations and models trained on the time-dependent simulation dataset.

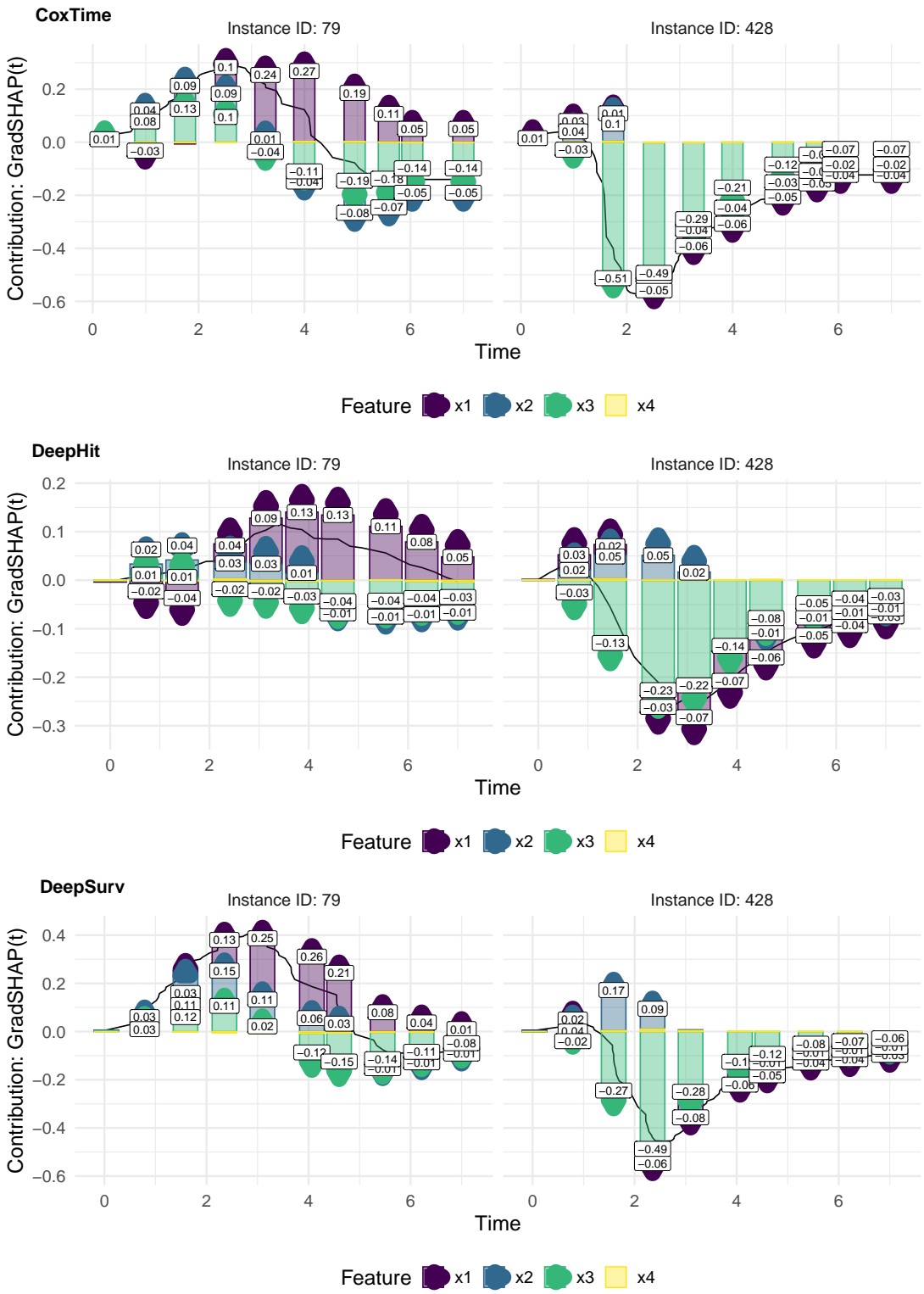

*Figure A.28.* GradSHAP(t) force plots for the selected observations and models trained on the time-dependent simulation dataset.

## A.3. GradSHAP(t) vs SurvSHAP(t)

### A.3.1. LOCAL ACCURACY

The concept of "local accuracy" originates from Shapley values and refers to the property that individual feature contributions sum up to the difference between the prediction and the average (i.e., marginal) prediction (Lundberg & Lee, 2017):

$$\sum_{j=1}^{p} \phi_j = f(\boldsymbol{x}) - \mathbb{E}_{\tilde{\boldsymbol{x}}}[f(\tilde{\boldsymbol{x}})]. \tag{11}$$

However, in the survival context, this decomposition must be considered for each time point, where the decomposition quantity dynamically changes over time since survival functions are monotonically decreasing. Krzyziński et al. propose a time-dependent variation of this measure to account for these dynamics:

$$M(t) = \sqrt{\frac{\mathbb{E}_{\boldsymbol{x}}\left[\left(f(t|\boldsymbol{x}) - \mathbb{E}_{\tilde{\boldsymbol{x}}}[f(t|\tilde{\boldsymbol{x}})] - \sum_{j=1}^{p} R_j(t|\boldsymbol{x})\right)^2\right]}{\mathbb{E}_{\boldsymbol{x}}\left[f(t|\boldsymbol{x})\right]}}. \tag{12}$$

This formulation gives greater weight to discrepancies at time points where the decomposition target becomes negligibly small, thus addressing situations where standard local accuracy would be less informative.

In our simulation study, we follow a similar setup as described for the time-independent effects (see Sec. A.1). We use $p = 20$ features with coefficients linearly increasing from 0 to 1 and alternating signs, i.e., $\beta_j = (-1)^j \frac{j-1}{19}$, and artificially censor the event time at 10. The data is split into a training set (1,000 observations) and a test set (100 observations). We train DeepSurv, CoxTime, and DeepHit models with two dense layers and 32 hidden nodes on the training data, using 30% validation data and early stopping for up to 500 epochs. For GradSHAP(t), we vary the number of integration samples (5, 25, 50). Additionally, for both XAI methods, we apply the methods on all 100 test instances. The results for all model classes are displayed in Fig. A.29. Further details can be found in the code supplement on GitHub.

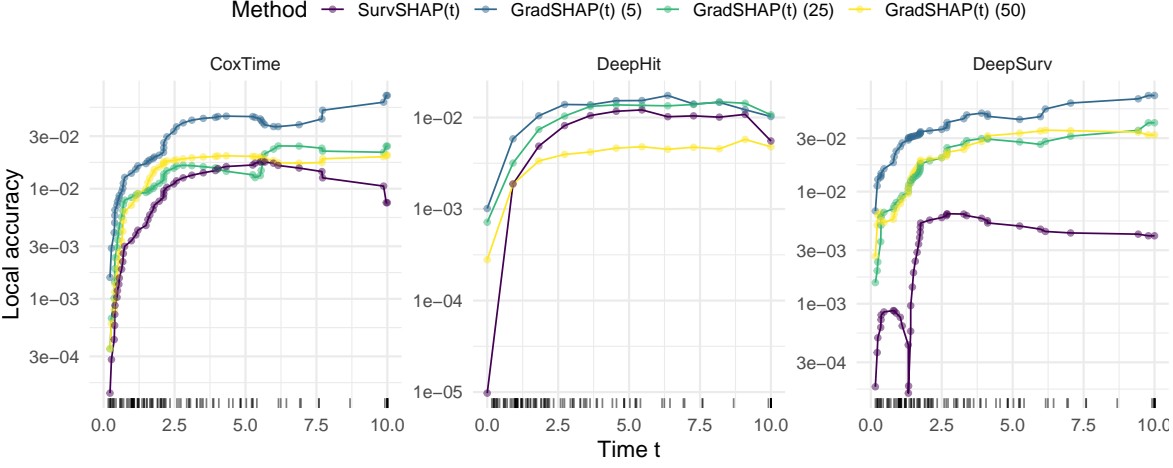

*Figure A.29.* Local accuracy (y-axis), measured as the normalized standard deviation of the difference between the black-box model output and the explanation (lower is better), plotted over time (x-axis) for SurvSHAP(t) (purple) and GradSHAP(t) with varying numbers of integration samples (blue = 5, turquoise = 25, yellow = 50). GradSHAP(t) achieves local accuracy comparable to SurvSHAP(t) while significantly reducing runtime.

### A.3.2. RUNTIME

To evaluate the computational efficiency of the proposed feature attribution methods, we compare SurvSHAP(t) and GradSHAP(t) with varying numbers of integration samples (5, 25, 50) across CoxTime, DeepHit, and DeepSurv models. The survival data is generated analog to the previous section but with varying $p$ and split the data into a training (1,000 observations) and a test set (100 observations). We train DeepSurv, CoxTime, and DeepHit models with two dense layers

and 32 hidden nodes on the training data, using 30% validation split and early stopping for up to 500 epochs. To obtain stable time measurements, we take the median runtime of 20 repetitions for a single trained model and repeat this five times. Additionally, we apply the XAI methods on all available test instances. To ensure a fair comparison of runtime performance, we do not employ any parallelization beyond controlling the number of threads. Specifically, we limit the number of `torch` threads and inter-op threads to 10 each. Further details can be found in the code supplement on GitHub.

Fig. A.30 illustrates the runtime (y-axis) as a function of the number of features (x-axis). The results reveal notable differences in computational demands between the methods. SurvSHAP(t) consistently exhibits higher runtime across all feature set sizes and model classes, with the computational cost rapidly increasing as the number of features grows. This can be attributed to its reliance on sampling multiple subsets of features for Shapley value estimation. In contrast, GradSHAP(t) demonstrates significantly improved efficiency, particularly when using fewer integration samples. As the number of features increases, GradSHAP(t) maintains computational efficiency, outperforming SurvSHAP(t) in all configurations.

Notably, increasing the number of integration samples from 5 to 50 for GradSHAP(t) results in higher runtimes but remains computationally more efficient than SurvSHAP(t) even for large feature sets. These findings underscore the scalability and efficiency advantages of gradient-based attribution methods for survival models when handling high-dimensional feature spaces.

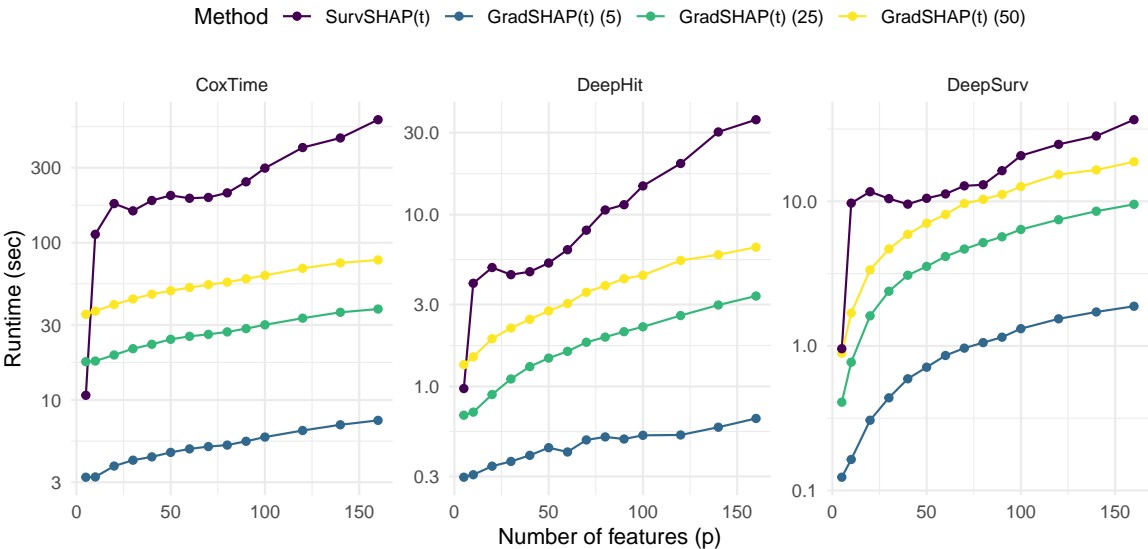

*Figure A.30.* Runtime (y-axis) for generating attributions using SurvSHAP(t) (purple) and GradSHAP(t) with varying numbers of integration samples (blue = 5, turquoise = 25, yellow = 50), measured on simulated datasets with an increasing number of features (x-axis). GradSHAP(t) significantly improves computational efficiency as the number of features grows, outperforming SurvSHAP(t) in all settings. For smaller feature sets, GradSHAP(t) with low integration samples also achieves faster runtimes than SurvSHAP(t). The results are averaged over multiple runs (20 per model).

### A.3.3. GLOBAL FEATURE RANKINGS

To assess the ability of the attribution methods to capture global feature importance, we evaluate GradSHAP(t), SurvSHAP(t), and SurvLIME on a simulated dataset with five features of predefined importance ($x_1 < x_2 < x_3 < x_4 < x_5$). The survival data is generated analogously to the runtime analysis with $p = 5$, $2{,}000$ train and $300$ test samples, and models (CoxTime, DeepHit, DeepSurv) are trained as described in the previous section. Further details can be found in the code supplement on GitHub.

Fig. A.31 presents the ranking distribution of features across the test set's 300 predictions for each model. Rankings are derived from the relative importance scores (i.e., the absolute values) assigned to each feature. Both GradSHAP(t) and SurvSHAP(t) demonstrate robust performance by consistently maintaining the predefined importance hierarchy across all models. This consistency highlights their ability to capture the underlying feature relationships. In contrast, SurvLIME shows a more uniform distribution of rankings, leading to a reduced capacity to differentiate features of varying importance.

Overall, the results emphasize that gradient-based methods like GradSHAP(t) and sample-based methods like SurvSHAP(t) are well-suited for identifying global feature importance in survival models, whereas SurvLIME may lack the precision required for nuanced analyses.

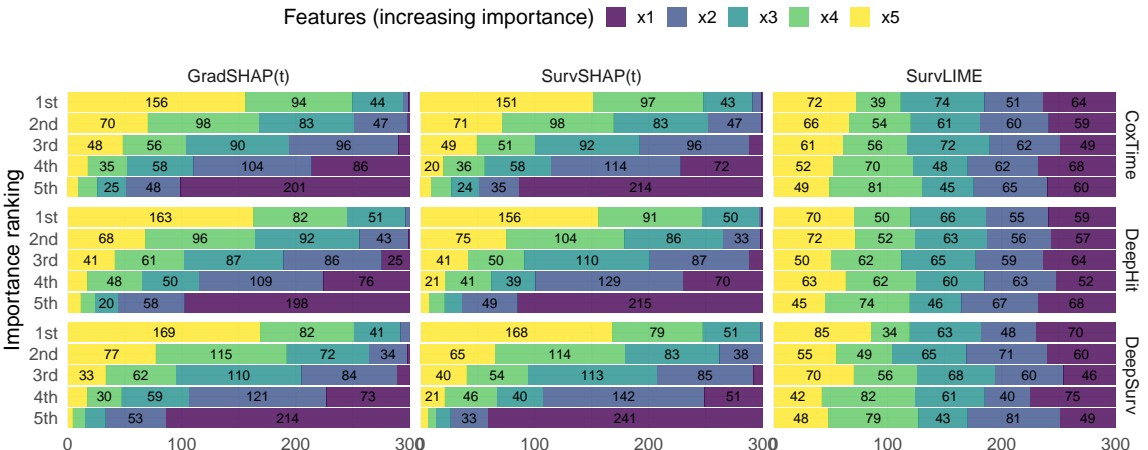

*Figure A.31.* Comparison of local and global importance rankings for 300 predictions from the CoxTime model (top), DeepHit model (middle) and DeepSurv model (bottom) on a simulated dataset including five features of increasing importance ($x_1 < x_2 < x_3 < x_4 < x_5$). Colors arranged in a gradient from purple to yellow reflect the global ranking of features within each model. GradSHAP(t) and SurvSHAP(t) perform similarly, consistently retaining the majority of observations for each consecutive feature. This demonstrates superior performance compared to SurvLIME, which produces more uniformly distributed rankings.

## A.4. Computational Details

A 64-bit Linux platform running Ubuntu 22.04 LTS with two AMD EPYC Genoa 9534 64-Core Processors (128 cores, 256 threads total), 1.5 terabytes RAM, and eight NVIDIA RTX 6000 Ada Generation GPUs (each with 48 GB memory) was used for all computations.

