# OpenReview forum: "Gradient-based Explanations for Deep Learning Survival Models"
_ICML.cc/2025/Conference — ICML 2025 poster_

### Official Review · Reviewer_5uUz · 2025-03-10

**Overall Recommendation:** 1

**Summary:**

The paper benchmarks previously proposed gradient-based explanation methods across three previously proposed deep survival analysis methods. Experimental results on synthetic and real-world datasets highlight differences in performance across scenarios.

**Claims And Evidence:**

- The paper claims an extension of previously proposed GradSHAP to GradSHAP(t) as a key contribution. However, it seems like a straightforward application of GradSHAP to survival outcomes. It is unclear what the actual contributions or challenges unique to survival time GradSHAP(t) address.
- The paper claims that GradSHAP(t) offers a better balance in computational efficiency and accuracy. However, only Figure 6 is provided to support this claim, and the definition of local accuracy is not provided in the paper. Additionally, it is unclear why alternative approaches such as SurvLIME are not discussed.
- The paper claims that gradient-based explanation methods effectively identify prediction-relevant features. Unfortunately, most experiments are based on synthetic data, with only Grad(t) and Grad(t) × Input(t) benchmarked on preselected instances. It is unclear why comprehensive evaluations of all gradient-based approaches, including GradSHAP(t), are not provided, with evaluations summarized across all instances.

**Essential References Not Discussed:**

- The paper should also discuss other non-time-varying survival explanation approaches, e.g., Kovalev et al. (2021) and Utkin et al. (2022).

**Experimental Designs Or Analyses:**

- Given that this is a benchmarking paper, the experimental results are underwhelming. The experimental comparisons are not consistent and seem cherry-picked. I encourage the author(s) to provide extensive analysis and results across all experiments, where all methods are included and results are summarized across all instances.

**Methods And Evaluation Criteria:**

- The proposed approach, GradSHAP(t), is a straightforward extension of the previously proposed GradSHAP to survival outcomes.
- The evaluation criteria on synthetic data is based on two preselected instances, which do not comprehensively capture the variance across test examples. Additionally, it is unclear what local accuracy in survival outcomes entails (Figure 6).
- The comparisons are not consistent across the deep survival methods and the gradient-based approaches across experimental settings.

**Other Comments Or Suggestions:**

**Minor**
- Eqn 4: Should be $\ln$ instead of $\log$

**Other Strengths And Weaknesses:**

The writing could be improved to focus on key contributions. Also, the experimental section is difficult to follow given the inconsistency in the experimental setup.

**Questions For Authors:**

- Could you provide comprehensive results for all experiments?

**Relation To Broader Scientific Literature:**

- Gradient-based explanations for deep survival models are an important research area for clinical decision-making

**Theoretical Claims:**

N/A

---

> ### Author Rebuttal · Authors · 2025-03-31
>
> Thank you for your valuable and insightful feedback. Before addressing your concerns in detail, we want to clarify a few crucial aspects and potential misunderstandings:
> * This is **not** a benchmark paper.
> * We do provide the definition and explanation of local accuracy in the appendix.
> * We do provide comprehensive results (as far as meaningful) for **all experiments** and **all methods** and for **aggregated** metrics.
> * We do compare with SurvLIME (contrary to the point you raised).
>
> **R4A1) What is the objective of this paper?**
>
> The objective of our paper is explicitly stated to **not** be a benchmark paper comparing different gradient-based explanation methods for survival DNNs (see XAI benchmark papers by Liu et al., 2021 and Agarwal et al., 2022 for classic prediction models). Instead, we adapt these methods for survival DNNs, provide method-specific visualizations and interpretations (addressing the recent disagreement problem), and compare GradSHAP(t) as a flexible model-specific version to SurvSHAP(t), making the calculation of Shapley values for survival DNNs possible (see **R1A3 and R3A1**).
>
> **R4A2) Why no comprehensive (aggregated) evaluations for all experiments? Only synthetic data?**
>
> A comprehensive evaluation across all methods is not feasible because XAI methods pursue different goals and lack definitive "correct" explanations. For gradient-based methods, this disagreement problem (Sturmfels et al., 2020; Krishna et al., 2023) origins from varying implicit or explicit baselines, making direct comparisons on a local level unreliable (e.g., Grad(t) measures output-sensitivity, GradxInput(t) attributes implicitly against a zero baseline, etc.). We evaluate the methods as local explanation techniques and highlight their characteristics using the introduced visualizations (see Appendix A.1 and A.2 for our comprehensive results). A meaningful aggregation of results across all instances is only possible for Shapley-based methods (as we did for two measures in Sec. 5.2). However, aggregation compromises the local nature of the explanations as it leads to global measures.
>
>  Additionally, we mainly used synthetic data for the comprehensive evaluation since it is the only reliable way to verify – in a controlled environment – if the model identifies truly prediction-relevant local effects based on the method's goal. Instead, a benchmark would focus on comparing methods against each other.
>
> **R4A3) Marginal technical contribution?**
>
> Please refer to the detailed response for **Reviewer 3 R3A1)**.
>
> We acknowledge that we may not have effectively communicated our intended contributions (adoptions for survival including challenges, visualizations and package implementation). In the final paper, we will emphasize and explain these contributions better.
>
> **R4A4) Concerns about local accuracy and advantage of GradSHAP(t)**
>
> The local accuracy measure for survival outcomes is defined and explained in detail in Appendix A.3.1. We acknowledge the importance of this metric and will incorporate it into the main text. It measures the average decomposition quality of the local attributions of SHAP-based methods (i.e., decomposing $\hat{S}(t|x)-E(\hat{S}(t|x))$). The primary advantage of our approach (GradSHAP(t)) lies in its drastically improved runtime efficiency compared to SurvSHAP(t). In addition to the results provided in our paper in Sec. 5.2. Fig. 6 and Appendix A.3.2., we now conducted an additional runtime comparison between SurvSHAP(t) and GradSHAP(t) as discussed in responses to **Reviewer 1: R1A1 and R1A3)**.
>
> **R4A5) Comparison to SurvLIME and other non time-varying methods.**
>
> We compared GradSHAP(t) with SurvLIME in terms of global feature importance ranking (see Sec. 5.2, Fig. 7, and A.3.3). While SurvLIME estimates local feature importance values, SurvSHAP(t) and GradSHAP(t) provide local attributions. A direct comparison is therefore only meaningful by evaluating the resulting feature ranking across all instances.
>
> In the related work, we purposefully only explicitly mention survival XAI methods, which are comparable to the gradient-based explanations discussed in the paper. Other survival explainability methods which do not provide time-dependent, local feature attributions are beyond the scope of our paper, including Kovalev et al. (2021) and Utkin et al. (2022), but are discussed in great detail in the referenced comprehensive review by Langbein et al. (2024).

---

> > ### Comment · Reviewer_5uUz · 2025-04-04
> >
> > Thank you for your rebuttal and for clarifying some of my concerns. Unfortunately, most of my concerns remain unaddressed, namely:
> >
> >  **Marginal Technical Contribution**
> >
> > GradSHAP(t) appears to be a straightforward extension of GradSHAP to survival analysis. Could you clarify what the technical contributions of the paper are?
> >
> >  **Underwhelming Experimental Results**
> >
> > Given the limited technical contributions, I would expect the experimental results to be rigorous enough to justify the paper's acceptance. However, there are several issues:
> >
> > 1) The results seem cherry-picked in terms of examples, survival models, and gradient-based explanation methods (Figures 3–8). The non-time varying methods should be included as well, since the estimate should just be constant over time.  Also the plots are difficult to interpret. Given than the effect of the covariates is known for synthentic data, an agregated quantitavie metric, e,g., RMSE comparing different methods should also be provided.
> >
> > 2) Given that SurvSHAP(t) is a comparable baseline to the proposed approach, it is unclear why SurvSHAP(t) is not consistently benchmarked against GradSHAP(t) in all these instances.
> >
> > 3) *Local Accuracy*: Thank you for pointing me to the definition. Could you clarify why local accuracy is directly tied to the specific gradient explanation method used?
> >
> > 4)  *Global Importance Ranking Tasks (Figure 7)*: The paper should also benchmark against non-gradient-based methods, such as the Cox proportional hazards model, to provide a more comprehensive comparison. Additionally, could you clarify the expected ground-truth feature ranking? Unfortunately, the proposed plot is difficult to interpret without actual ground-truth information. This is another instance where summarizing the results with a quantitative metric would be helpful.

---

> > > ### Author Response · Authors · 2025-04-08
> > >
> > > Thank you for your time and effort in reviewing our responses. We regret that some of your concerns remain unaddressed, and we appreciate the opportunity to clarify these further.
> > >
> > > **Marginal Technical Contribution?**
> > >
> > > As already noted in our rebuttal, we have addressed this point in the response to Reviewer 3 under **R3A1** and kindly refer you to that response, where we provide a detailed explanation.
> > >
> > > **Underwhelming Experimental Results?**
> > >
> > > **1) Cherry-picked examples**
> > >
> > > * Our paper focuses on **local post-hoc** attribution methods, specifically adapting gradient-based methods for survival neural networks (SNN). While this choice may appear selective, it covers the most common SNN and gradient-based attribution methods, and thus clearly defines the paper's scope. Additionally, since the paper is about local gradient-based methods, it is essential to show the results instance-wise (Fig. 3-5), even though it seems cherry-picked, but is the nature of the methods discussed in the paper.
> > > * Including experiments on non-post-hoc or non-attribution methods (e.g., inherent explanations or counterfactuals) would blur our contribution scope as we explain a single prediction of an already trained SNN and not (directly) the survival data. While the mentioned non-time-varying methods may be of interest in the broader context of survival XAI, they do not align with our focus on post-hoc feature attribution methods and thus fall outside the scope of our detailed examples.
> > > * In our simulations, we aimed to showcase the different local behavior of the methods and highlight the "Disagreement Problem" in the survival context, which can only be effectively demonstrated in a simulated setting on an instance-wise level. We acknowledge that this motivation may not have been made clear enough in the current version and will improve this in the final revision.
> > > * As already mentioned, the methods pursue different decompositional objectives – also depending on the baselines – we are unsure how the methods could be meaningfully compared using RMSE, but would gratefully appreciate clarification. While correlation-based comparisons are possible, such analyses have already been performed for standard models. Instead, we compare methods where a shared objective and model-agnostic counterparts exist, such as GradSHAP(t) and SurvSHAP(t).
> > >
> > > **2) SurvSHAP(t)} vs. GradSHAP(t)**
> > >
> > > It is unclear which instances you are referring to where GradSHAP(t) was not compared to SurvSHAP(t). Sec. 5.1 is not intended as a benchmark, but rather as a proof of concept to demonstrate that explanations align with model-learned feature effects and provide guidance for correct interpretation. Sec. 5.2 then uses an equivalent data-generating process to benchmark GradSHAP(t) and SurvSHAP(t) – the only directly comparable methods – **across all instances in the simulated datasets** using three global evaluation metrics: local accuracy, runtime, and global importance ranking.
> > >
> > > **3) Local Accuracy**
> > >
> > > We use the time-dependent local accuracy criterion $M:T\to R_{>0}$ (plotted over the survival time $t$):
> > >
> > > $$
> > > M(t)=\sqrt{\frac{E_{x}\left[\left(f(t|x)-E_{\tilde{x}}[f(t|\tilde{x})]-\sum_{j=1}^p R_j(t |x)\right)^2\right]}{E_{x}\left[f(t|x)\right]}}.
> > > $$
> > >
> > > Both GradSHAP(t) and SurvSHAP(t) are (marginal) Shapley-based attribution methods, aiming to decompose the difference between a single and the expected survival predictions $f(t|x)-E_{\tilde{x}}[f(t|\tilde{x})]$, thus quantifying feature contributions (see Fig. 2 and Sec. 4.2). The other gradient-based methods instead:
> > >
> > > * Grad(t) and SG(t) are output-sensitivity methods (no decomposition goal),
> > > * Grad x Input(t) decomposes $\approx f(t|x)$,
> > > * IG(t) decomposes $f(t|x) - f(t|\tilde{x})$.
> > >
> > > Only for IG(t) exist mathematical guarantees for an exact approximation, i.e., an equivalent *local accuracy* measure could be defined. However, there is no (implemented) model-agnostic counterpart for IG(t).
> > >
> > > **4) Global importance rankings**
> > >
> > > The ground truth feature ranking is given by the feature indices ($x_1<x_2<x_3<x_4<x_5$) as highlighted in the plot legend ("Features (increasing importance)") and discussed in Sec. 5.2. We will further clarify the feature order in description and legend of Fig. 7 in the final version. We agree that a summary metric would improve clarity, so we will compute and include the rank correlation between the ground truth and observed feature rankings. While we can also fit a CoxPH model and compute its FIs $\beta_j x_j^{(i)}$, the focus of our study is to compare XAI methods for NN-based models. Introducing a CoxPH comparison would shift the goal to evaluating model quality rather than the performance of XAI methods, which is beyond the scope of this paper. As discussed in **R4A4**, SurvLIME, SurvSHAP(t), and GradSHAP(t) are the only relevant local XAI methods for survival analysis in this context.

---

### Official Review · Reviewer_dEe2 · 2025-03-14

**Overall Recommendation:** 3

**Summary:**

This paper shows a comparative study on various explanation methods for survival analysis.  While there are several model-agnostic methods to interpret models for survival analysis, this paper considers gradient-based methods.  The applicability of the gradient-based methods is limited to models that can compute gradients (e.g., neural network models), but this paper shows its effectiveness compared to the other methods.

**Claims And Evidence:**

Yes.

**Essential References Not Discussed:**

None.

**Experimental Designs Or Analyses:**

Yes.

**Methods And Evaluation Criteria:**

Yes.

**Other Comments Or Suggestions:**

None.

**Other Strengths And Weaknesses:**

A weakness of this paper is that the technical contribution is marginal.  While there are many methods to interpret $y=f(x)$ for the standard regression analysis where $x$ is a feature vector and $y$ is a target value, this paper shows only adaptations of these methods to interpret $p=S(t|x)$ where $t$ is a time point and $p$ is a probability and the adaptations are almost straightforward.  In other words, this paper does not show any novel idea associated with applying the interpretation methods for the standard regression analysis to survival analysis.  (This is in contrast with (Krzyzinski et al., 2023), which proposes a modification of an evaluation metric specifically designed for survival analysis.  The modification is described in the appendix of this paper: from Equation (9) for the standard regression analysis to Equation (10) for survival analysis.)

A strength of this paper can be seen as explicitly providing the adaptations (as summarized in Figure 2) and shows effectiveness of these adapted methods.   The results reported in the experiments (in Section 5) are reasonable and convincing.  For example, this paper shows the effectiveness of using the gradient-based method, GradShap(t), compared with existing model-agnostic methods SurvSHAP(t) and SurvLIME.

**Questions For Authors:**

None.

**Relation To Broader Scientific Literature:**

While Langbein et al. (2024) review many methods to interpret survival models, gradient-based methods are briefly discussed in this review.  This paper focuses more on gradient-based methods, and the contribution of this paper can be seen as extensive experiments on comparative study of gradient-based methods for survival models.

**Theoretical Claims:**

No theoretical claims are presented in this paper.

---

> ### Author Rebuttal · Authors · 2025-03-31
>
> Thank you for your careful evaluation and suggestions. We acknowledge that our contributions may not have been communicated clearly enough in the original submission. To address this, we will revise the manuscript to better clarify these key contributions:
>
> * Our work follows adaptations common in survival XAI but tackles key challenges beyond straightforward mathematical extensions. E.g., a naive application of these methods on CoxTime is not possible.
> * We introduce tailored visualizations and post-hoc interpretations for gradient-based survival outputs and exemplify the impact of implicit vs. explicit baselines, addressing the recent debate on the disagreement of gradient-based explanations.
> * As a novel contribution, we provide a software package implementing all described gradient-based XAI methods for DeepSurv, DeepHit, and CoxTime.
>
> In the following, we provide a detailed explanation regarding your feedback:
>
> **R3A1)  Marginal technical contribution?**
>
> Our primary contribution is extending six standard gradient-based explanation methods to time-dependent survival analysis, addressing a crucial gap in survival XAI research. The extensions are far from trivial from both a technical implementation and post-hoc interpretation perspective. For instance, in the CoxTime model time is an input feature, allowing for complex relationships between the artificially created time feature and other features. Thus, applying gradient-based methods naively to the survival function $S(t|x)$ separately at each time point $t$ is not feasible, as time-expanded instances are no longer independent, leading to accumulated gradients from earlier time points. This computational difficulty is not captured in the formal mathematical adaptions. Similar methodological adaptations are standard practice in survival interpretability research (see Kovalev et al., (2020), Krzyzinski et al., (2023); even the time-dependent local accuracy metric in Krzyzinski et al., (2023) follows a similar extension from the original Shapley value axiom.
>
> Furthermore, another major contribution of our work are effective visualization and interpretation techniques for functional outputs tailored to different methods. This is particularly important given the ongoing debate and disagreement regarding gradient-based methods (Sturmfels et al., 2020; Krishna et al., 2023; Koenen et al., 2024). Our work contributes to this discussion by clarifying how implicit or explicit baselines in these methods influence survival explanations and provides practical guidance on selecting appropriate techniques based on their interpretability characteristics.
>
> Finally, we highlight our `R` package, `Survinng`, as an additional contribution. Existing libraries like `Captum` and `innsight` do not natively support survival DNNs, necessitating custom implementations. Our package, which includes all described gradient-based XAI methods for DeepSurv, DeepHit, and CoxTime, will be made available with the final version of this paper.

---

### Official Review · Reviewer_xZMF · 2025-03-16

**Overall Recommendation:** 4

**Summary:**

The authors introduce GradSHAP(t), an extension of SurvSHAP(t) that analyzes the gradients to better explain the model’s predictions. The authors also propose extensions of other gradient-focused XAI methods to align with the survival task.

**Claims And Evidence:**

Yes.

**Essential References Not Discussed:**

While the paper considers GradSHAP(t) with respect to proportional hazard (DeepSurv, CoxTime) and discrete time (DeepSurv) models, I would be interested in an additional analysis of Accelerated Failure Time (AFT) model such as DART (Lee et al, 2023).

**Experimental Designs Or Analyses:**

This article relies on synthetic data for a bulk of its experiments. However, given that real world data is often difficult to explain (due to lack of domain knowledge), this is necessary for the purpose of this paper. These results are reinforced by a single real-world dataset, which contains known features of interest.

The experimental setup is well documented. Also, the motivation for the design decisions made when constructing the synthetic dataset are clearly stated and understandable.

The authors provide rigorous and convincing analysis of all experiments.

**Methods And Evaluation Criteria:**

GradSHAP(t) (and the other proposed methods) clearly align with the application at hand. Explainability of survival models is critical due to their applications in fields such as healthcare. The temporal aspect creates additional complexity, which motivates the exploration of time-dependent explainability analysis.

**Other Comments Or Suggestions:**

N/A

**Other Strengths And Weaknesses:**

The extension of gradient based XAI to survival analysis is novel and has clear potential for future impact. The article is overall written well and all figures are visually digestible.

**Questions For Authors:**

N/A

**Relation To Broader Scientific Literature:**

The paper is related to both the survival analysis literature and the explainability research literature.

**Theoretical Claims:**

The authors do not provide theoretical claims.

---

> ### Author Rebuttal · Authors · 2025-03-31
>
> We appreciate the suggestion to include semiparametric AFT-based survival deep learning models, such as Deep AFT Rank-regression for Time-to-event prediction model (DART). It is an interesting approach, which estimates the survival function in similar fashion to a non-Cox-based version of the DeepSurv model. However, the pre-trained baseline hazard function additionally depends on the output of the base neural network, as highlighted in Eq. 9 in the paper by Lee et al. (2023). Unlike Cox-based methods, such as DeepSurv and CoxTime, this formulation allows for gradient computation on the baseline hazard, which could provide further insights into the differences between Cox and non-Cox methodologies. Given its potential, we aim to explore its integration and explanations in future research.
>
> Our current focus is on methods that are readily accessible to practitioners, particularly those implemented in widely used libraries such as Pycox (Python) and survivalmodels (R). Expanding these packages or developing more comprehensive survival analysis software to also include AFT models such as DART is an important direction, and we are actively working toward addressing this gap.
>
> More broadly, extending gradient-based XAI methods to AFT-based deep learning models like DART is an exciting avenue for future work. We will highlight this in the paper's future work section.
>
> We would also like to take the opportunity to highlight that we conducted additional comparisons of SurvSHAP(t) and GradSHAP(t) on real data. In the multi-modal model (Sec. 5.3), SurvSHAP(t) was aborted after 10 hours (256 threads, 700GB RAM) with only two reference samples, while GradSHAP(t) completed in ~8 minutes using 100 reference and 20 integration samples. See our response to Reviewer 1 (**R1A3**) for details.
>
> Thank you very much for the positive feedback!

---

### Official Review · Reviewer_xEYK · 2025-03-17

**Overall Recommendation:** 3

**Summary:**

This paper addresses the challenge of interpreting "black box" deep learning models used for survival analysis, which predict time-to-event outcomes. The authors introduce a framework for gradient-based explanation methods to capture the time-dependent influence of various features, including those from multi-modal data like medical images and tabular information. They introduce GradSHAP(t), a gradient-based, model-specific counterpart to the model-agnostic SurvSHAP(t). Using both synthetic and real-data experiments, it is shown to be computationally efficient while maintaining accuracy compared to existing approaches.

**Claims And Evidence:**

The primary claims for novel contributions in the paper are
1. using existing gradient-based methods to explain survival predictions (specifically, capturing the time-dependence of features)
2. introducing GradShap(t) and showing that it is computationally efficient while maintaining accuracy with its main competitor SurvShap(t) from a previous work

Prior work has already established the importance time-dependent attribution of features to survival predictions. Claim 1 extends this specifically for gradient-based explanations.

For Claim 2, the evidence seems preliminary. We seem to lose in terms of the local accuracy (Fig 6). Moreover, I'm not sure how important the corresponding gains in runtime even are, since this is a post-hoc task, not an inference task where runtime is critical. The in-depth evaluation is done on 2 examples presented in Fig 6, and I'd have liked to additionally see aggregate metrics over the dataset. In the end, I'm left thinking it might provide marginal benefits over the existing method SurvShap(t), if at all.

**Essential References Not Discussed:**

N/A

**Experimental Designs Or Analyses:**

I have pointed out the issues with the evaluations used, especially for the real-world data. I think the manuscript should contain not just 2 in-depth examples but also aggregate metrics over the dataset. It should also clearly compare with SurvShap(t) on the same examples and identify where the gains of GradShap(t) are coming from. As a reviewer it is hard for me ascertain based on the limited results here whether GradShap(t) works significantly better and what its benefits over SurvShap(t) are.

**Methods And Evaluation Criteria:**

- The synthetic experiments make sense in that they give us an idea of potential benefits and pave the way for real-data experiments.
- The real-data experiments are lacking in that the gains relative to the main baseline, SurvShap(t), are unclear. The authors chose 2 examples but even on those, I see gains in runtime (which is not too important for a post-hoc task) but a loss in local accuracy (which is arguably more important for interpretation). I am also unable to parse Fig. 8 to judge whether what the model tells us is indeed sensible.

**Other Comments Or Suggestions:**

N/A

**Other Strengths And Weaknesses:**

Strengths: I really enjoyed reading the paper, it is very well written.

Weaknesses: I'd like to see more of
- explanation of SurvShap(t) and how exactly GradShap(t) gains relative to it
- experiments with real data that illustrate the above

**Questions For Authors:**

N/A

**Relation To Broader Scientific Literature:**

The work extends interpretability of survival models from model-agnostic methods to gradient-based methods. The hope here is that while model-agnostic methods only deal with the model outputs, the gradient-based methods use the model's "internals" to get a better view into how the model is using its features.

**Theoretical Claims:**

N/A

---

> ### Author Rebuttal · Authors · 2025-03-31
>
> We are grateful for your constructive feedback. In response, we conducted additional experiments on the feasibility and computational efficiency of GradSHAP(t) and SurvSHAP(t) on the multi-modal real data example, which we will include in the final paper. Here, GradSHAP(t) took 5 minutes to compute, while we had to abort the computation of SurvSHAP(t) after 10 hours (details below). In light of these results, we argue that runtime is one of the most crucial aspects for post-hoc Shapley value approximation, which is supported by previous literature. Additionally, we want to highlight that local accuracy and importance ranking are already aggregated metrics. The following summarizes all your concerns in-depth:
>
> **R1A1) Limited gain of GradSHAP(t) relative to SurvSHAP(t)?**
>
> The objective behind GradSHAP(t) is not merely to propose a "better" method, but rather a practical one that balances flexibility and computational feasibility. Although differences in local accuracy are visible due to the log scale, they are practically negligible, whereas the runtime improvement of GradSHAP(t) is substantial. This becomes particularly apparent in our additional experiments, including image data, which demonstrate that SurvSHAP(t) quickly becomes computationally infeasible without substantial resources, whereas GradSHAP(t) remains efficient and can be computed on a standard laptop (for further details refer to **R1A3**). Computational runtime is a well-established criterion in post-hoc XAI methods, particularly for SHAP explanations. It is frequently used as a key selling point in the scientific literature when introducing new estimation methods, see e.g., TreeSHAP (Lundberg et al., 2020), FastSHAP (Jethani et al., 2021) or various other SHAP algorithms such as in Ancona et al. (2019) or Chen et al. (2018) and runtime is included as a comparison metric in their SHAP benchmarking suite (Python SHAP package, Lundberg and Lee, 2017).
>
> **R1A2) Limited number of examples/lack of aggregated metrics?**
>
> Our comparison of SurvSHAP(t) and GradSHAP(t) was performed for all three survival DNN classes (see Appendix A.3 for full plot) with varying numbers of input features, averaged over 20 trained models per case. In spite of its name, local accuracy is already a dataset-wide aggregated measure (see Eq. 10, Appendix A.3.1.), which we plot for $p=30$ over the survival time in Fig. 6. Additionally, Figure 7 presents a comparison of global feature importance, particularly importance rankings, which are also aggregated over survival time. Could you clarify if you were referring to something beyond these analyses?
>
> **R1A3) Real data?**
>
> We performed additional comparisons of SurvSHAP(t) and GradSHAP(t) on real data. On the multi-modal model (including images) for explaining a single instance (Sec. 5.3), we had to abort the computation of SurvSHAP(t) after 10 hours (using 256 threads and 700GB of RAM) with only two reference samples. In contrast, GradSHAP(t) completes in around 8 minutes with 100 reference and 20 integration samples. As an additional comparison, we conducted the same experiment (but ResNet18) on downscaled images (32×32) using a standard ML workstation (48 threads, 256GB RAM). As shown in the table below, SurvSHAP (with 50 samples) takes more than 41 times longer than GradSHAP (n = 25, samples = 50), which is nearly 25 minutes for just a single explanation compared to 36 seconds. GradSHAP achieves this not only faster but also with better instance-wise local accuracy (averaged over survival time $t$, i.e., no aggregation over all instances).
>
> We agree that this is crucial information for the readers and we will include discussions and visualizations of the computational efficiency of both methods on real data in the final version of the paper.
>
> | Method                                 | Runtime           | "Instance local accuracy" (avg. over $t$) |
> |----------------------------------------|-------------------|---------------------------------|
> | GradSHAP(t) (n = 10, samples = 10) | 2.96 sec                     | 0.00108                         |
> | GradSHAP(t) (n = 25, samples = 50) | 36.07 sec                   | **0.00023**                         |
> | GradSHAP(t) (n = 50, samples = 50) | 1 min 18.05 sec         | **0.00021**                    |
> | SurvSHAP(t) (samples = 5)                | 2 min 45.30 sec         | 0.00230                         |
> | SurvSHAP(t) (samples = 25)              | 12 min 34.78 sec       | 0.00044                         |
> | SurvSHAP(t) (samples = 50)              | 24 min 45.93 sec       | 0.00027                         |
>
>
> We are also happy to provide further clarification on Figure 8, if you further specify which aspects remain unclear from the figure and/or text.

---

### Decision · Program_Chairs · 2025-05-01

**Decision:**

Accept (poster)

**Comment:**

After the discussion period, three reviewers favored acceptance (2 weak accept, 1 accept) and one favored rejection. In looking at the discussion, I would say that the authors have largely addressed the reviewer concerns, and the dissenting reviewer's concerns have been adequately addressed (in my opinion) by the authors' two responses to this reviewer. I get the impression that one of the major issues with the originally submitted draft was a lack of clarity of technical contributions, which was sufficiently addressed in the discussion period (see author comment "R3A1"); this sort of explanation really needs to be incorporated into the paper and made clear as early as possible. At the moment, I get the impression that some of the added results during the discussion period and some of the clarifications would definitely improve the paper draft but also constitute somewhat significant paper edits that arguably might warrant another round of reviewing. I am recommending a cautious "weak accept".